# From Words To Rewards:
# Leveraging Natural Language For Reinforcement Learning

**Belén Martín-Urcelay** *burcelay3@gatech.edu*
*Department of Electrical and Computer Engineering*
*Georgia Institute of Technology*

**Andreas Krause** *krausea@ethz.ch*
*Department of Computer Science*
*ETH Zurich*

**Giorgia Ramponi** *giorgia.ramponi@uzh.ch*
*Department of Computer Science*
*University of Zurich*

**Reviewed on OpenReview:** *https://openreview.net/forum?id=GbxOpLANdf*

## Abstract

We explore the use of natural language to specify rewards in Reinforcement Learning with Human Feedback (RLHF). Unlike traditional approaches that rely on simplistic preference feedback, we harness Large Language Models (LLMs) to translate rich text feedback into state-level labels for training a reward model. Our empirical studies with human participants demonstrate that our method accurately approximates the reward function and achieves significant performance gains with fewer interactions than baseline methods.

## 1 Introduction

Reinforcement Learning (RL) (Sutton & Barto, 2018) is a powerful framework for solving complex decision-making problems by training agents to maximize cumulative rewards through interactions with an environment. RL has achieved remarkable success in a variety of domains, from games (Mnih et al., 2015) and robotics (Kaufmann et al., 2023) to healthcare (Yu et al., 2021) and finance (Pendharkar, 2022). Central to the RL paradigm is the concept of a reward function, which provides the agent with feedback on its actions and guides its learning process.

Defining a suitable reward function in real-world applications is often difficult or impractical (Hadfield-Menell et al., 2017), limiting the deployment of RL in scenarios where desired behavior is hard to specify precisely. To address these challenges, Reinforcement Learning from Human Feedback (RLHF) (Christiano et al., 2017) has emerged as a promising approach. Rather than relying on predefined reward functions, RLHF derives a reward signal directly from human input, ensuring it better reflects human values and intentions. This strategy has been especially effective in domains where human judgment is crucial for determining task success.

While most RLHF approaches rely on comparing or ranking trajectories, humans naturally communicate intent through more nuanced textual descriptions (Cherry, 1966). Shifting from comparison-based feedback to textual input would allow for a richer expression of underlying goals (Metz et al., 2024). However, for such text-based feedback to be useful in RL, it must be effectively translated into a suitable reward model for planning. In this paper, we introduce a novel approach for learning reward models from natural language feedback. We bridge this gap by employing Large Language Models (LLMs) to map expressive human text into state-level labels, which we use to update a reward model. This allows the reward model to capture

the context and subtleties of human preferences more effectively, leading to more robust and adaptable RL agents. Our contributions are as follows:

- We propose an in-context learning approach using LLMs to map natural language feedback into labeled state-level examples for training a reward model.
- We incorporate our reward modeling approach into an RLHF framework and validate its performance in a Gridworld environment through experiments with 26 human participants and in continuous environments.
- We show that human feedback, especially when provided with guidelines, can proactively steer agents toward unexplored high-reward states, avoiding common reward modeling challenges.
- Our method achieves strong performance with as few as 10 human interactions, outperforming preference-based RL. In two environments (Gridworld and Rubik's cube), we additionally observe settings where it outperforms ground-truth reward baselines.

By harnessing human insights from textual comments, our method aims to bridge the gap between human preferences and machine learning, paving the way for more adaptable and human-aligned RL.

## 2 Related Work

### 2.1 Preference Based Reinforcement Learning

Traditional RL relies on explicit reward functions to drive the learning process. When the reward function is not known or difficult to construct, RLHF suggests we collect human feedback to model the reward. In Preference Based Reinforcement Learning (PbRL) (Busa-Fekete et al., 2014; Christiano et al., 2017), human oracles provide their preferences between pairs of trajectories. These preferences are used to train a reward model, enabling the deployment of standard RL algorithms to find the optimal policy. Unfortunately, as depicted in Figure 1, relying solely on comparisons misses out on valuable information about finer details of the reward (Basu et al., 2018; Peng et al., 2024). We propose using natural language feedback to overcome this limitation.

### 2.2 Learning from Natural Human Feedback

A natural way for humans to interact and express their intentions is through text. Consequently, there is much interest in leveraging natural language in RL. One common strategy is to map natural language instructions to trajectories or features. To achieve this mapping, previous works limit the instructions to a finite set (Goyal et al., 2019; Bahdanau et al., 2019; Nguyen et al., 2021; Lin et al., 2022), or force a specific sentence structure, e.g., "Go to X" (Fu et al., 2019). These restrictions simplify the mapping process, but, unlike our algorithm, they also limit the flexibility of the language used. Another approach, which allows for more general language, employs Neural Networks (NN) to map from natural language to rewards (MacGlashan et al., 2015; Tung et al., 2018; Narasimhan et al., 2018; Yang et al., 2021). While this approach achieves good performance, it requires a large labeled dataset to train the NNs. To avoid curating extensive datasets, recent research focuses on exploiting pre-trained models. Pre-trained valence analyzers (Hutto & Gilbert, 2014) translate text feedback into a sentiment score. At each iteration, the sentiment score of the whole text drives the Bayesian update of the reward model (Sumers et al., 2021). A single sentiment score may not

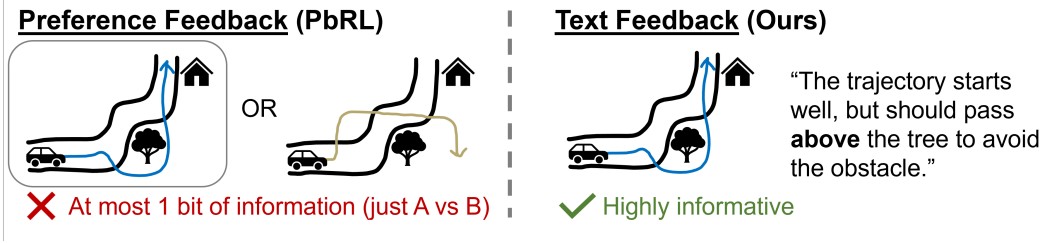

Figure 1: Traditional RLHF uses binary preferences, providing just one bit per query. We propose natural language feedback, offering richer, human-readable signals that reduce interaction needs.

capture all nuances in human text, e.g., "*the start is good, but the end is bad.*" Our experiments show how having a more fine-grained interpretation of the feedback is beneficial.

### 2.3  Language Models in Reinforcement Learning

LLMs have recently emerged as powerful tools for natural language processing, providing novel approaches in the field of RL. One direction treats pretrained LLMs as proxies for reward signals, directly querying whether an outcome satisfies a language description of an objective (Kwon et al., 2023). While promising, this binary feedback is limited in expressivity, an inherent information bottleneck that our method aims to overcome. Another approach harnesses LLMs' general knowledge to provide common sense priors (Li et al., 2023; Ahn et al., 2022; Peng et al., 2024) that bias agent actions, for example, identifying hazardous states from metadata (Choi et al., 2022). These methods complement our approach, which aims to learn subjective human intentions that are unknown to LLMs.

A third line of work employs LLMs to translate natural language descriptions of goals into reward functions in the form of code (Zhu et al., 2025). This approach faces two key challenges: (1) it is often easier to criticize outcomes than to articulate goals precisely, leading to underspecified tasks, and (2) generating reward models from descriptions alone frequently results in reward misalignment, especially when intent is underspecified. To address these issues, researchers have developed iterative revision approaches using either pre-defined reward evaluation functions (Yu et al., 2023; Ma et al., 2024; Xie et al., 2023) or binary preference feedback (Yu et al., 2024a; Sun et al., 2025) to judge the performance of the generated reward functions. However, reward evaluation functions rarely exist in practice, and we demonstrate that binary feedback provides limited information. Furthermore, all these methods depend on LLM-generated code snippets which may be incorrect (Liu et al., 2024) and unreliable (Yu et al., 2024b).

Our approach takes a different route: we use LLMs to convert free-form human text feedback, which provides more nuanced evaluations than preference feedback, into training data for reward modeling. While there are a few experiments in which authors provide text feedback (Ma et al., 2024; Xie et al., 2023), we collect and analyze text feedback from a broad set of independent human participants, strengthening the real-world applicability of our findings. Additionally, instead of relying on LLM-generated diversity to avoid local maxima, our method uses text feedback to guide exploration towards relevant states.

## 3  Problem Setting

In this paper, we consider an *agent* who interacts with an *environment* aiming to maximize an expected reward. We describe the interactions between the agent and the environment as an episodic Markov Decision Process without reward function (MDP\R) (Puterman, 2014). Formally, an episodic Markov Decision Process without reward function (MDP\R) is a tuple $\mathcal{M} := (\mathcal{S}, \mathcal{A}, \mathbb{P}, T)$. $\mathcal{S}$ is the state space where each state $\mathbf{s} \in \mathcal{S}$ captures the environment configuration. For example, the position of an agent in a grid, the position of a car and all surrounding objects in a self-driving task or the prompt and partial answer in LLM fine-tuning. $\mathcal{A}$ is the set of actions that the agent can perform in the environment; for example, an action could be a step to the right on the grid, accelerating a car or generating a specific next token in LLM fine-tuning. $\mathbb{P} : \mathcal{S} \times \mathcal{A} \to \Delta(\mathcal{S})$ captures the transition probabilities, mapping state-action pairs to a probability distribution of the next state over $\mathcal{S}$, where $\Delta(\mathcal{S})$ denotes the probability simplex over $\mathcal{S}$, and $T$ is the time horizon. $\mathbb{P}$ captures the environment dynamics, such as the distribution of a car's position after acceleration or how text evolves after token generation. The reward function, which maps state-action pairs to a reward $r : \mathcal{S} \times \mathcal{A} \to \mathbb{R}$, is unknown to the agent. Instead, the agent learns a reward model $\widehat{r} : \mathcal{S} \to \mathbb{R}$ based on human feedback. The unknown reward function reflects how well state-action pairs align with the human's implicit preferences or goals.

At each step, the agent performs an action according to a *policy* $\pi : \mathcal{S} \to \mathcal{A}$. The goal is to learn the policy $\pi^*$ that maximizes the expected return from the current state $\mathbf{s}$,

$$V_r^t(\mathbf{s}) = \max_{\mathbf{a} \in \mathcal{A}} r(\mathbf{s}, \mathbf{a}) + \sum_{\mathbf{s}' \in \mathcal{S}} \mathbb{P}(\mathbf{s}'|\mathbf{s}, \mathbf{a}) V_r^{t-1}(\mathbf{s}'), \tag{1}$$

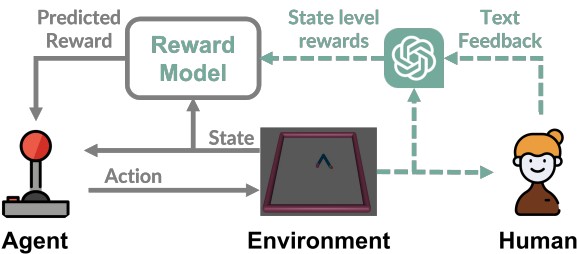

Figure 2: Block diagram of Reinforcement Learning from Human Text Feedback (RLHTF). The algorithm consists of two iterative phases: (1) learning a reward model (dashed lines) from state-level labels derived by an LLM from evaluations in natural language, and (2) policy learning (solid lines), where an agent is trained using standard RL algorithms that query the learned reward model.

---

**Algorithm 1** RLHTF

---

1: **Inputs:** number of interactions $N$, landmarks, $f_{\mathrm{LLM}}$
2: Initialize Policy $\pi_0$ and reward model $\widehat{r}_0$
3: **for** $i = 0$ **to** $N - 1$ **do**
4:     Record trajectory following policy: $\mathbf{t}_i \leftarrow \pi_i$
5:     Query human for feedback: $\mathbf{f}_i \leftarrow \mathbf{t}_i$
6:     Encode context: $u_i \leftarrow (\mathbf{f}_i, \mathbf{t}_i, \text{landmarks})$
7:     Translate to state-reward pairs: $\{\mathbf{s}_o, R\} \leftarrow f_{\mathrm{LLM}}(u_i)$
8:     Update reward model: $\widehat{r}_{i+1} \leftarrow \text{reward\_update}(\widehat{r}_i, \{\mathbf{s}_o, R\})$
9:     Update policy: $\pi_{i+1} \leftarrow \text{policy\_update}(\pi_i, \widehat{r}_{i+1})$
10: **end for**

---

where $V_r^1(\mathbf{s}) = \max_{\mathbf{a} \in \mathcal{A}} r(\mathbf{s}, \mathbf{a})$, $t \in [1; T]$ represents the number of timesteps left until the trajectory finishes, and $s'$ represents the next possible state based on the environment's transition dynamics. We denote the state-action pairs visited when following $\pi^*$ as the optimal trajectory $\boldsymbol{\tau}^* = \{\mathbf{s}_t^*, \mathbf{a}_t^*\}_{t=1}^T$.

We propose an algorithm, Reinforcement Learning from Human Text Feedback (RLHTF), shown in Figure 2. Analogously to the classical RLHF framework (Christiano et al., 2017), RLHTF consists of two phases that are executed iteratively: reward model learning and policy learning. Section 4 describes how the reward model $\widehat{r}$ is learned from human text feedback. Since the agent does not have direct access to the true reward function $r$, the agent instead learns the policy $\widehat{\pi}$ that maximizes the value function $V_{\widehat{r}}$ derived from the estimated reward model $\widehat{r}$. The agent directly queries the reward model, instead of the human evaluators, significantly reducing time, energy, and monetary costs during policy learning (Christiano et al., 2017). We employ standard RL methods. The algorithm is outlined in Algorithm 1 and publicly available[1].

Our primary goal is to minimize the performance gap between this learned policy $\widehat{\pi}$ and optimal policy $\pi^*$, which we denote as *value gap* and formally define as

$$\mathbb{E}\left[\sum_{t=1}^T r\left(\mathbf{s}_t^*, \mathbf{a}_t^*\right) - r\left(\widehat{\mathbf{s}}_t, \widehat{\mathbf{a}}_t\right)\right],$$

where $\{\widehat{\mathbf{s}}_t, \widehat{\mathbf{a}}_t\}_{t=1}^T$ represents the trajectory when following $\widehat{\pi}$.

## 4 Learning a Reward Model from Human Text Feedback with LLMs

We break down the process of learning a reward model from human feedback into three steps: (1) Context encoding, which gathers human feedback with environment information; (2) Translation function, which transforms the encoded context into a structured signal; (3) Reward model update mechanism to incorporate

---

[1]https://github.com/BelenMU/WordsToRewards

these signals. This decomposition has been shown to be effective in prior work (Metz et al., 2024). Here, we show how these steps can be implemented with text feedback.

## 4.1 Context Encoding

To encode raw data from the environment and interactions with evaluators into a format suitable for an LLM, we must construct a structured user prompt. This prompt should capture the agent's trajectory, human feedback, and any relevant environmental landmarks to provide full context for interpretation.

At each interaction, the agent follows the policy $\widehat{\pi}$, generating a *trajectory* $\{\widehat{\mathbf{s}}_t\}_{t=0}^{T}$. A human evaluator observes the trajectory and provides *text feedback* $\mathbf{f}$. These evaluations are flexible; for example, they may include criticisms of specific states ("the last step is horrible"), or suggestions for alternative, unexplored states ("go to the door"). The human text feedback is combined with information about the trajectory to create a user prompt

$$u = \{\text{feedback: } \mathbf{f}, \text{ trajectory: } \{\widehat{\mathbf{s}}_t\}_{t=0}^{T}, \text{ landmarks: } [(\text{name}_1, \text{location}_1), ..., (\text{name}_N, \text{location}_N)]\}.$$

As shown in Figure 6, the user prompt may also include information about *landmarks* in the environment that help ground human feedback. Namely, we may include landmark names and their locations in the environment, providing reference points known by both the user and the LLM.

## 4.2 Translation function

We use LLMs' language processing capabilities to transform the information in the user prompt into labeled states, which are then used to train a reward model. We design a *system prompt* to guide the LLM in its role as a translation function. Efficient system prompts must describe the LLM's role and the items in the user prompt (Schulhoff et al., 2024). As this context is environment dependent (e.g., the meaning of elements in the states), the system prompt differs from environment to environment. However, the system prompt remains consistent across all user interactions. **Appendix D includes all prompts verbatim.**

There are three key components that make our system prompts efficient. First, to enhance reasoning, we employ *chain of thought (CoT)* prompting (Wei et al., 2022), asking the model to classify the feedback intent (e.g., evaluation vs. correction) before identifying the relevant states. Feedback is inherently intent-dependent (Metz et al., 2024); for example, "to the left of the lamp is good" could either be an evaluation validating a past action or an instruction suggesting a future correction. This intermediate classification step helps the translation function better interpret and adapt to human intent, leading to more precise state-level label assignments. Second, we use *few-shot prompting* (Kaplan et al., 2020) and provide demonstrations to steer the model to better performance. By exposing the model to relevant cases, we reduce ambiguity and improve accuracy. Third, to ensure that the output of the LLM is reliably interpretable in downstream tasks, we enforce a structured format. Rather than relying on free-form text generation, which can be inconsistent, we employ *function calling* to guarantee a well-defined output, making it easier to identify relevant states and rewards.

Given the appropriate system prompt, the LLM translates the user prompt $u(\mathbf{f}, \{\widehat{\mathbf{s}}_t\}_{t=0}^{T}, \text{landmarks})$ into a labeled dataset of states $\{\mathbf{s}_o, R\} = f_{\text{LLM}}(u)$, where each output state $\mathbf{s}_o \in \mathcal{S}$ has a corresponding reward $R \in \mathbb{R}$. This dataset is used to train the reward model.

## 4.3 Reward Model Update

The goal is to learn a model of the reward conditioned on the state-reward pairs $\{\mathbf{s}_o, R\}$ output by the LLM. In simple tabular settings, the agent may track the reward probability distribution for every state and update it using Bayesian inference as the state-reward pairs are observed. In more complex or continuous environments, we may approximate the reward function with an NN. At each iteration, we expand the training dataset with the state-reward pairs generated by the LLM, and finetune the NN using supervised learning.

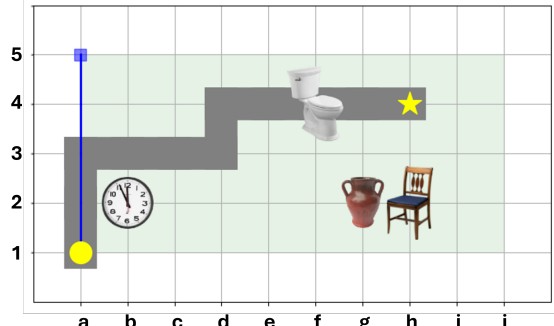

Please enter your critique of the trajectory:

**Answer 1**: "Steps 1 and 2 are correct. Go above the clock and then through the toilet." (Describes specific states.) — **RLHTF with instructions**
**Answer 2**: "Follow the gray path until you reach the star." (Uses unknown references, e.g., star) — **RLHTF no instructions**
**Answer 3**: "It is terrible." (Describes whole trajectory) — **Sentiment Feedback**

Figure 3: *Gridworld* environment. The aim of the agent is to follow a target path (in gray) from the start (yellow circle) to the end (yellow star). The agent's trajectory (in blue) currently deviates from this path. Below, we show the critique input box presented to human participants, along with example responses from different feedback conditions.

## 5 Experiments with Human Evaluators

We recruited 26 human participants to evaluate our approach[2]. In particular, we apply RLHTF to the Gridworld environment shown in Figure 3. An agent aims to follow a target path $\tau^*$ known to human evaluators but unknown to the agent. Standard RL approaches, like Goal Condition RL (GCRL), assume access to a reward function, which is precisely what our method aims to learn, making them inappropriate for comparison. Instead, we empirically compare against ground-truth feedback, which is rarely available in practice, and state-of-the-art approaches for RL with human feedback:

- **True trajectory-level feedback**: The agent receives a single ground truth accumulated reward for the entire trajectory, based on the number of steps that match the target path. This accumulated reward is uniformly applied to all states in the trajectory.
- **True state-level feedback**: The agent receives a ground truth reward for each state in the trajectory, indicating whether the state is on the target path.
- **Sentiment feedback**: Evaluators convey their judgment implicitly through the sentiment of their text feedback. For example, an evaluator may write "Very bad" for poor performance or "Amazing" when the agent closely follows the target path. The agent measures and uniformly applies the sentiment score as a reward across all states in the trajectory (Sumers et al., 2021).
- **PbRL**: Human evaluators observe two candidate trajectories and select the one they judge to be closer to the target (Christiano et al., 2017).

Experiments with true environment feedback were simulated 50 times. For the rest of the experiments, we followed A/B testing guidelines and randomly assigned evaluators to eight different target paths among two distinct feedback conditions. This resulted in approximately 50 experiments per feedback type. All experiments start with the same reward model prior and consist of 4 interactions, during which the agent executes the current optimal trajectory and receives the corresponding feedback to update the reward model. Further implementation details, experimental design and participant information are provided in Appendix B.1.

Since a state is either on the target path or not, in the RLHTF setting we prompt the LLMs to output binary rewards $R' \in \{0, 1\}$. All prompts are available verbatim in Appendix D. Additionally, to contextualize feedback, the environment includes randomly placed landmarks, such as a chair or a clock. These landmarks serve as reference points that both evaluators and LLMs observe, enabling the LLMs to translate feedback such as "*Go above the clock*" in Figure 3 into a positive reward at position 'b3'.

Once feedback is collected, the agent updates the reward model accordingly. The agent tracks the reward probability distribution for every state, which we model as a beta distribution. In all scenarios, we initialize

---

[2]The study was categorized as minimal risk research qualified for exemption by the Institutional Review Board (IRB).

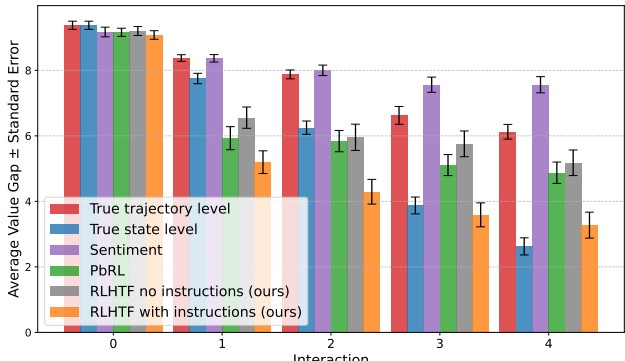

Figure 4: Performance comparison of algorithms in the *Gridworld*. When participants receive instructions on how to phrase their feedback, RLHTF outperforms other human feedback based methods and even surpasses true environment feedback in early iterations.

the reward distributions as $\beta(0.5, 0.5)$ to introduce a bias towards binary rewards: 0 (negative) or 1 (positive). Then the agent performs the trajectory that maximizes the current reward model. As the time horizon is finite ($T = 10$), we solve for the optimal trajectory with dynamic programming. The agent receives feedback based on the executed trajectory. For analytical tractability, we model the distribution of the observed rewards $R'$ with the conjugate prior, i.e., as Bernoulli distributions.

## 5.1 Feedback Granularity and Performance

Figure 4 shows the evolution of agent error across successive interactions with evaluators. Here, *error* is defined as the number of steps in which the agent deviates from the target path. Statistical significance after four interactions was assessed using Welch's t-test, as detailed in Appendix B.1.

All feedback methods lead to performance improvements, but the *time granularity* (Metz et al., 2024), i.e., the resolution at which feedback is provided (trajectory vs. state), plays a decisive role in learning efficiency. In our experiments, both the true trajectory-level reward and sentiment feedback operate at the *trajectory-level*, offering a single evaluation for an entire trajectory. This coarse granularity limits the agent's ability to discern which specific states contributed to success or failure, thereby slowing learning. In contrast, true state-level reward and RLHTF operate at *state-level*, providing more precise guidance during training. The impact of these granularity differences is particularly evident when comparing the first two bars in Figure 4: after four iterations, true state-level feedback yields less than half the error of true trajectory-level feedback. This statistically significant ($p < 10^{-15}$) improvement reinforces the advantage of detailed information about each state's contribution to accelerate learning. Similarly, our algorithm RLHTF (5th bar) outperforms the sentiment baseline (3rd bar) across all interactions, reducing the error by more than half when leveraging the fine-grained information in the text. Additionally, RLHTF remains competitive with the state-of-the-art PbRL algorithm, underscoring the effectiveness of $f_{\text{LLM}}$ in extracting actionable information from human text feedback.

## 5.2 Human Feedback Adaptability to Instructions

In our initial evaluation, we found that without instructions on how to construct the text feedback, participants often included information that the LLMs could not interpret. For example, when presented with Figure 3, one participant wrote: "follow the gray path until you reach the star." However, the LLM has no knowledge of the gray path or the star; in fact, this is precisely what the agent is attempting to learn. Participants may also refer to history not provided in the prompt ("This is not even as good as the initial one. Retract to your first attempt.") or actions outside of the reward model scope ("go up more on the right.").

To address this issue, we showed the evaluators the following list of five instructions on how to provide effective feedback:

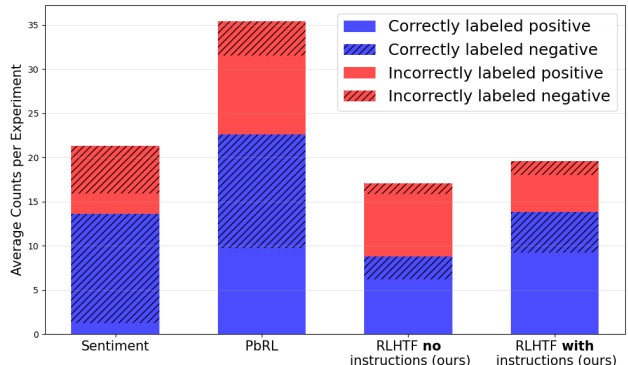

Figure 5: Average count of state-labels generated per experiment, categorized by correctness (correct in blue, incorrect in red) and predicted labels (positive in solid, negative in hatched). A state-label is correct if the label accurately identifies whether the state is in the target path or not. Providing brief instructions to human evaluators (RLHTF with instructions) substantially increases the proportion of correctly identified states.

---

**Additional Instructions**: The agent has limited capabilities, so for it to understand you correctly you should restrict your feedback. Namely, the agent does not understand:

- Do not compare trajectories: treat each path individually, without reference to previous attempts. For example, avoid feedback like: "Now it is worse, go back to the previous trajectory".
- Do not refer to the position of the star, yellow circle, or gray road. The agent doesn't know their locations; in fact, the agent is trying to learn where these are. For example, avoid feedback like: "Follow the road until the star".
- Avoid specific movement descriptions (go up, turn right). For example, **avoid feedback like: "go up, up, right, up" or "turn right later".**

Instead, the agent understands well:

- Description of states: Position in the trajectory. e.g. "At the beginning is wrong", "step number 6 is good", or position with respect to the furniture, e.g. "You should go to the left of the couch".
- Sentiment: It works especially well if you explain what states are good or bad. e.g. **"The first half of the trajectory is bad. Above the TV is good."**

---

We conducted 51 experiments in the RLHTF with instructions condition. Remarkably, providing participants with just two paragraphs of instructions increases the comprehensibility of the feedback to the LLM. For example, in the same setting of Figure 3 a participant wrote: "steps 1 and 2 are correct. Go above the clock and then through the toilet." This allows $f_{\text{LLM}}$ to better extract relevant spatial cues and assign rewards accordingly, as shown in Figure 5. We evaluate whether each state-reward pair, automatically generated from human feedback, is correct by comparing it against the known ground truth path. Formally, a generated state-reward pair $(s_o, R)$ is considered *correct* if $R = \mathbb{1}\{s_o \in \tau^*\}$, where $\mathbb{1}\{\cdot\}$ is the indicator function, and $\tau^*$ is the ground truth target path. Providing brief instructions to the evaluators increases the accuracy of LLM-generated state-labels by 19.1%, indicating that the same amount of human effort yields substantially clearer, more actionable supervision. In contrast, the sentiment baseline receives mostly negative feedback and attains only 35.4% positive precision on the states in $\tau^*$. PbRL assigns rewards to every differing state across a pair of trajectories, which inflates label counts with redundant or weakly informative items and results in a 6.9% lower label accuracy than RLHTF with instructions (63.7% vs. 70.6%). Moreover, RLHTF with instructions encourages evaluators to provide more constructive guidance: it is the only condition in which the amount of positive feedback surpasses negative (1.25x). These trends indicate that instructions help steer feedback, improving both reward attribution and, consequently, policy success rates. Moreover, as detailed in Appendix B.1, providing guidelines significantly improves consensus: it doubles the agreement between human evaluators' feedback interpretations and increases the consistency of interpretations across different LLMss.

After a single instance of feedback, Figure 4 shows that RLHTF with instructions reduces the task error by 42%, even outperforming ground-truth state-level feedback. This improvement occurs because RLHTF proactively guides exploration towards the high-reward regions mentioned by the human feedback, which might not be covered by the current trajectory. In contrast, true environment feedback is purely reactive, offering evaluations only for states actually visited. Although ground-truth state feedback eventually surpasses RLHTF after 4 interactions, our method outperforms other RL algorithms without explicit rewards. A key advantage of RLHTF is its use of LLMs to perform reward attribution. Unlike previous approaches, which update all states in a trajectory (sentiment) or all differing states between two trajectories (PbRL) with a single reward, RLHTF distributes rewards more precisely. This precision allows RLHTF to reduce error more quickly and substantially, significantly outperforming *Sentiment* ($p < 10^{-13}$) and *PbRL* ($p < 0.0025$) algorithms. After just 4 instances of human feedback, RLHTF with instructions reduces the error to approximately one-third of its original value.

Beyond task error reduction, Table 1 shows the *success rate*, defined as the percentage of experiments in which the learned reward model, after four pieces of feedback, leads to a policy that exactly follows the target path. Strikingly, RLHTF with instructions matches the success rate of true state-level feedback, achieving a 25.5% success rate. This level of performance is particularly impressive given the combinatorially large space of possible trajectories and the fact that only four pieces of feedback are provided. In contrast, baseline methods (true trajectory-level feedback, sentiment feedback, and PbRL) fail to produce a single perfect path. This result demonstrates that guided human feedback, when processed by RLHTF, can rival access to privileged ground-truth state rewards in our experimental setting.

| Algorithm | Success Rate |
|---|:---:|
| True trajectory-level | 0% |
| True state-level | 25.0% |
| Sentiment (Sumers et al., 2021) | 0% |
| PbRL (Christiano et al., 2017) | 0% |
| RLHTF no instructions (ours) | 0% |
| **RLHTF with instructions (ours)** | **25.5%** |

Table 1: Percentage of learned policies that perfectly follow the target path after four rounds of feedback. RLHTF with instructions matches the performance of true state-level feedback, while all other methods fail to achieve any success.

As LLMs become more powerful, their ability to process human feedback improves, further boosting the performance of RLHTF in the Gridworld experiments. As shown in Appendix B.1, upgrading from GPT-4o (used in our main experiments) to the newer GPT-4.5 or Sonar-reasoning models further reduces the error by 23% and 30%, respectively. This result suggests that, unlike the baselines, RLHTF in settings similar to those tested here may continue to benefit from future advances in LLM capabilities, further strengthening the potential of natural language feedback in RL.

## 6 RLHTF in Continuous and Structurally Complex Environments

In the previous section, we demonstrated that RLHTF efficiently learns reward functions in discrete, tabular domains. Here we analyze method's performance in two settings that are *continuous* and *structurally complex*. We focus on two representative challenges: (i) precise motor control in the MuJoCo *Reacher* task, where rewards must be inferred from a continuous observation space, and (ii) goal–conditioned manipulation in a Rubik's Cube environment, where the reward hinges on matching human-specified color patterns. Across these experiments, we compare RLHTF to preference-based RL (PbRL) and to an oracle with access to the true reward.

## 6.1 Generalization to Continuous Environments

To test generalization beyond tabular settings, we apply RLHTF in a continuous environment of the physics simulator MuJoCo (Todorov et al., 2012). Namely, we apply RLHTF to *Reacher*, a two-jointed robot arm. The aim is to apply appropriate torques to the hinges so that the robot's fingertip reaches a target. The human evaluators watch and critique a video of the robot arm moving under the current policy. To aid human evaluators in their assessment, we incorporate a timestamp and some visual landmarks (colored circles) into the environment. Figure 6 shows the modified Reacher environment. These modifications aid the human evaluator to refer to specific locations (*"Go to the left of the blue circle."*) or moments in the trajectory (*"The movement from frame 4 to 8 is wrong."*). We emphasize that our method does not require the human to explicitly understand or annotate states such as joint angles or torques. Instead, human evaluators provide natural language feedback, and our method infers meaningful state-level reward signals from contextual and temporal cues naturally present in the text.

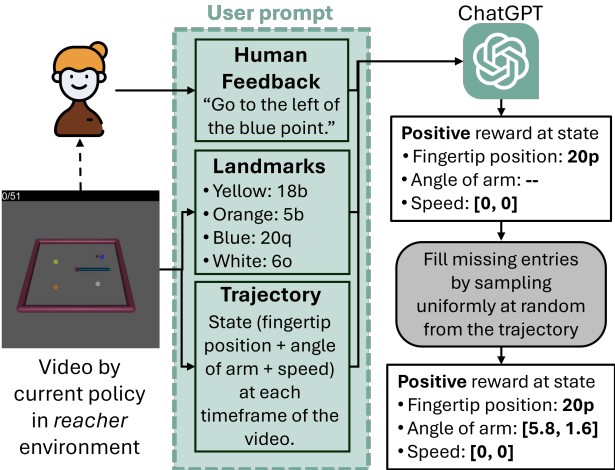

Figure 6: Generation of state-reward pairs from human feedback and observations using an LLM (Step 7 in Algorithm 1) in modified *Reacher* environment. The aim of the robotic arm is to stay close to the red ball.

The LLM processes three inputs: human feedback, landmark locations and the sequence of filtered states (where information about the target is removed, see Appendix B.2), making up the video. The LLM's task is to interpret these inputs to deduce what states are described as positive or negative. Figure 6 shows how the state-reward pairs are generated. The LLM outputs pairs of filtered states and binary labels. Each label indicates whether the corresponding state is positive or negative. If a filtered state is returned partially, missing elements are filled in by randomly sampling from the set of observed trajectory states. We approximate the reward function as a fully connected NN, which takes a state vector as input and outputs a scalar corresponding to the predicted reward. At each iteration, we expand the training dataset with the state-reward pairs output by the LLM, $\{\mathbf{s}_o, R\}$. Then, we employ stochastic gradient descent to fine-tune the NN with the expanded training dataset. This supervised learning process uses a cross-entropy loss function and is further detailed in Appendix B.2.

We suggest that feedback in natural language is more informative than preferences between two trajectories, and thus RLHTF requires fewer interactions with humans to achieve an accurate reward model. To verify the hypothesis, we compare the evolution of the reward model with RLHTF versus PbRL, as shown in Figure 7. While both feedback types improve the reward model, text feedback results in a more precise localization of the target. By the tenth interaction, only a small area around the target receives a high reward with RLHTF, whereas PbRL results in a reward model more uncertain about the target location, giving high reward to large areas of the environment. Figure 7 suggests that natural language feedback enables the reward model to identify the target more quickly and precisely.

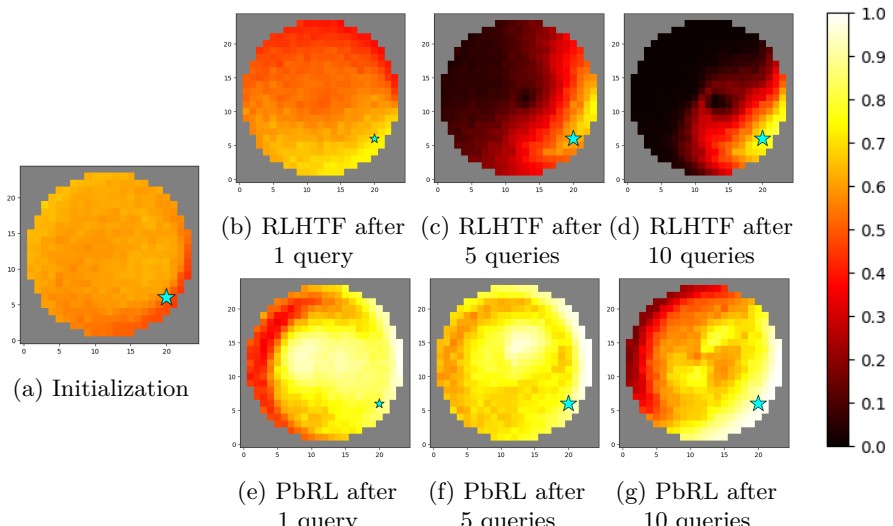

(a) Initialization

(b) RLHTF after 1 query

(c) RLHTF after 5 queries

(d) RLHTF after 10 queries

(e) PbRL after 1 query

(f) PbRL after 5 queries

(g) PbRL after 10 queries

Figure 7: Reward model visualization for RLHTF vs PbRL. The top row illustrates the evolution of the reward estimations as more text feedback is gathered, while the bottom row shows the corresponding evolution with preference feedback. The blue star marks the target location and darker colors indicate lower predicted rewards. Notably, RLHTF quickly converges to a more accurate reward model than PbRL, as evidenced by the more localized high-reward region around the target.

We also compare the agent's performance, in terms of average distance to the target, when there is a budget of 10 human interactions. Figure 8a shows how the reward evolves as the agent learns. Although RLHTF is not as effective as directly observing the true environment feedback, RLHTF performs much better in regimes with low feedback than PbRL. In fact, in our experiments RLHTF increases the reward by 40% with only 10 human inputs, while PbRL at first repeatedly executes the trajectory with the arm fully bent, and it needs many more pairwise comparisons to approach the target.

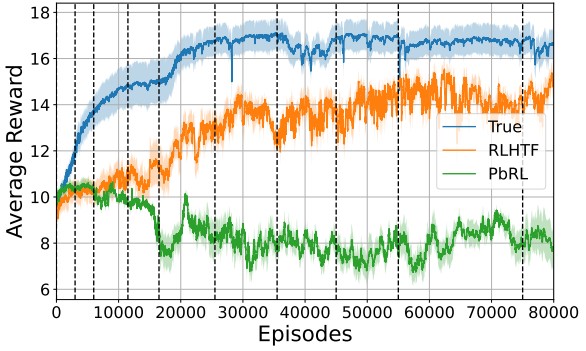

(a) Reacher environment. RLHTF demonstrates strong performance in continuous environments, increasing the reward by 40% with only 10 human inputs.

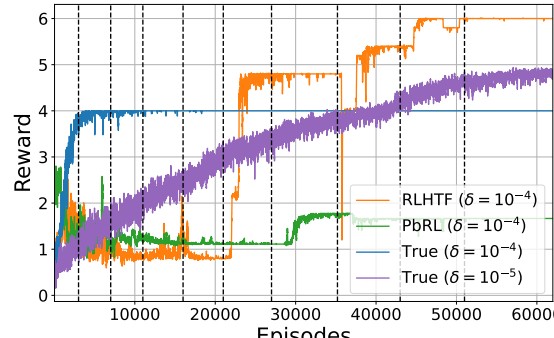

(b) Rubik environment for task where the optimal reward is 7.4. Unlike true reward, which can lead to local maxima, RLHTF leverages dynamic human feedback that adapts to agent progress, guiding exploration and overcoming reward modeling challenges.

Figure 8: Reward evolution with episodes of the REINFORCE algorithm. We compare the performance for RLHTF with 10 pieces of text feedback provided at the vertical dashed lines, PbRL with 10 trajectory comparisons, and true environment.

### 6.2 Towards Adressing Certain Reward Modeling Challenges

To highlight RLHTF's strengths in environments where reward design is challenging, we evaluate RLHTF in the Rubik's Cube environment (Gröling, 2022). This environment involves a six-sided cube where each side has a $3 \times 3$ grid of squares, each taking one of six colors. The goal of the agent is to manipulate the cube so that the front face matches a pattern specified by a human user.

Consider the task of obtaining an orange 'X' pattern on the front face of the Rubik's cube, i.e., both diagonals must be orange. In RLHTF, the evaluator watches a video of the trajectory following the current agent policy, and then provides text feedback. This feedback, along with the sequence of nine color grid squares observed on the front face, is input to the LLM. The LLM processes this information and outputs sets of 9 color grid squares, each accompanied by a label indicating whether the set is positive or negative. The labeled dataset is then used to update the reward model. In the case of PbRL, we query the evaluator for their preference between two different videos. The candidate videos are chosen by sampling trajectories and identifying the pair with the highest variance in preference estimations among the reward models in an ensemble. The true reward is computed by assigning a +2 score for each correct orange square in the diagonals and averaging this score over the ten timesteps in each trajectory.

Figure 8b compares the performance of RLHTF, PbRL, and true reward. After only ten human interactions, RLHTF successfully produces the desired orange 'X' pattern on the Rubik's Cube front face. In contrast, PbRL shows no meaningful improvement beyond its baseline performance. This limitation arises because preference-based feedback encodes at most one bit of information per interaction, requiring many more interactions to achieve the desired performance. Using true rewards, the agent gets stuck in a local maximum, where only three out of the five diagonal squares are correctly orange. Even after manipulating the learning rate, the agent fails to produce the desired pattern. This shows how reward design in RL problems is challenging and often leads to unwanted behavior, even in constrained environments. In contrast, evaluators in RLHTF naturally adapt their feedback according to the agent behavior, mitigating such misalignments. For example, an evaluator may write "you are doing it wrong, the top right corner should be orange and not red", thus encouraging exploration when stuck in a local maximum. This adaptability bypasses the need for complex reward design and extensive hyperparameter tuning. Human text feedback and additional experiments with a different target pattern are provided in Appendix B.3.

## 7 Future Work

While our framework introduces a new paradigm for incorporating natural language human feedback into reinforcement learning, its current implementation presents several limitations. First, our prompts must be adapted to each environment, similar to how traditional RL requires handcrafting reward functions. Although we demonstrate some degree of generalizability across three distinct environments, fully task-agnostic prompting remains an open challenge. The system prompts require manual design to ground the LLM in the specific context of each environment. In our experiments, we partially reduce this burden by using off-the-shelf object detectors (i.e., YOLO) to automatically detect and name landmarks, which are then inserted into the prompt. However, the core system prompt still requires manual design. To further automate the system prompt generation, future work could leverage metadata (e.g., simulator documentation or object annotations) to automatically generate descriptions or relevant items, (e.g. the description of state elements). Alternatively, future work could explore automatic prompt optimization techniques (Ramnath et al., 2025), that use evolutionary strategies to discover effective prompts without human intervention. We focus on three environments chosen for their interpretability and relevance to key RL challenges. Future work will extend our approach to more complex tasks, such as LLM finetuning, where an evaluator could critique stylistic elements ("too formal") or specific content ("the third paragraph is confusing"), going beyond the binary preferences typically used in existing RLHF pipelines.

Second, although LLMs can interpret rich feedback, their performance is constrained by the content and clarity of the prompt. When critical information is missing or ambiguous, the LLM may hallucinate or misinterpret the intent. We find that providing instructions to human evaluators significantly reduces these issues, though it does not eliminate them entirely. Multimodal verification could help reduce hallucations further. Future work

could deploy an ensemble method, where each piece of human feedback is interpreted by several models, and only outputs with majority agreement are retained. Our analysis suggests that keeping only the state-reward pairs on which the majority of models agree will filter many of the incorrect interpretations. Similarly, our method shows consistent empirical improvements, but we lack formal guarantees of performance.

Lastly, our experiments focus on state-level feedback. However, our approach naturally extends to action-level feedback by appropriately prompting the LLM, e.g., an evaluator on the Gridworld environment might say "go up 5 steps." This capability opens the door to broader applications in RL settings.

## 8 Conclusions

Our work leverages LLMs to extract state-level rewards from natural language feedback, addressing the challenge of reward attribution and capturing more nuanced information than simple binary comparisons or a single sentiment score. Moreover, our experiments with real human participants, a contribution not common in this line of research, take a step toward real-world applicability.

### Acknowledgments

Belén Martín-Urcelay was supported by the Rafael del Pino Fellowship. The human-participant experiments were supported by the Hasler Stiftung grant (project 2024-06-12-90). The authors thank the action editor and the anonymous reviewers for their insightful comments and constructive feedback.

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

# A Broader Impact Statement

## A.1 Human Subject Research

Our work involves a study with human participants. The study was categorized as minimal risk research qualified for exemption status under 45 CFR 46 104d.2 by the Institutional Review Board (IRB). All 26 participants were adults, provided informed consent before participation, were able to withdraw at any time, and were equally compensated $10 for approximately 30 minutes of their time (no volunteer work was used). No personally identifying information beyond coarse demographic data was collected, stored or shared.

The ultimate goal of our work is to reduce the need for complex reward engineering by enabling alignment by non-experts from their feedback in the form of natural language. This new manner of alignment may contribute to the democratization of RL training. However, as with all alignment technologies, there is a dual-use risk where such methods could be employed to align agents with malicious instructions. Malicious actors could exploit natural language feedback interfaces to inject misleading rewards, causing RL agents to learn harmful behaviors. All our experiments were done in simulated environments with no deployment to real-world systems or high-stakes decision domains. Any future applications in safety-critical or socially sensitive settings would require additional domain-specific oversight and safeguards.

The proposed algorithm seeks to align RL agents with fewer human interactions. However, employing fewer evaluators for alignment may exacerbate bias. This reinforces the importance of hiring diverse evaluators across cultural, gender and language backgrounds.

## A.2 Reliance on LLMs

This work uses LLMs as translation functions to map human feedback onto state-level rewards. However, LLMs are prone to hallucinations or biased interpretations, especially when presented with ambiguous feedback. We attempt to mitigate these risks by using few-shot prompting, environment-specific prompts and providing evaluators with guidelines, however, misinterpretations still persist. Moreover, the performance of our approach depends on proprietary LLM models, which may limit transparency and reproducibility. To partially mitigate this, we release our prompts and code, and report results across multiple LLMs.

# B Experiment Details

## B.1 Gridworld

This section provides detailed information about the experimental setup, data collection process, and implementation choices for our experiments with human evaluators. In each trajectory, the agent takes 10 steps in the *Gridworld* environment shown in Figure 3. At each step, the possible actions are either to move right or up. The agent movements are restricted to a 5x10 grid. When the agent is at a border and performs an action that would take the agent outside of the allowed region, the agent does not move.

We simulate a scenario in which a human wants a robot to navigate their house in a specific manner. To make the environment more realistic, we sample four images of furniture or household objects (e.g., sofa, chair, toaster and TV) and add them to the environment. These objects serve as landmarks, helping evaluators communicate their critiques more effectively by using them as reference points. The landmarks are detected with a YOLOv8 model (Jocher et al., 2022) trained on the Microsoft COCO dataset (Lin et al., 2014), and their positions are fed to the LLM for shared context with the evaluators. Namely, we use *gpt-4o* to translate human feedback into state-level rewards.

We conducted our experiments with the assistance of human evaluators. The study was categorized as minimal risk research qualified for exemption status under 45 CFR 46 104d.2 by the Institutional Review Board (IRB). We recruited 26 participants, most of whom were university students (46% female, 54% male). Table 2 provides a breakdown of their geographic region were participants were born.

Following A/B testing guidelines, we randomly assigned evaluators to interact with different algorithms. Each evaluator provided feedback on two out of the four algorithms: (1) either RLHTF with instructions or RLHTF

| Region | Participant Count |
|---|:---:|
| Asia | 11 |
| North America | 6 |
| Africa | 2 |
| South America | 1 |
| Europe | 1 |

Table 2: Breakdown of participants by region of birth.

without instructions, and (2) either Sentiment or PbRL. To minimize any potential bias due to familiarity with the interface or tasks, we randomized the order of these settings. Each participant provided feedback for either 8 different rooms or for a duration of 30 minutes, whichever occurred first. Participants were compensated with $10 for their work. In total, we collected 772 feedback samples from 26 participants. The average number of words per interaction is 17.48 and 17.30 words in the settings without and with additional guidelines, respectively. A breakdown of the number of experiments for each algorithm using human feedback is summarized in Table 3. The experiments with the true environment rewards were simulated 50 times.

| Experiment Type | First | Second | Total |
|---|:---:|:---:|:---:|
| RLHTF with instructions | 27 | 24 | 51 |
| RLHTF without instructions | 23 | 22 | 45 |
| PbRL | 24 | 32 | 56 |
| Sentiment | 24 | 17 | 41 |

Table 3: Breakdown of number of experiments for each algorithm type, each experiment was done for four interactions. Columns *First* and *Second* indicate whether the feedback was collected in the first or second task performed by the evaluator, with totals shown in the last column.

We conduct Welch's t-test to compare the performance of different algorithms after receiving four pieces of feedback. Table 4 reports the resulting $p$-values. Our results indicate that RLHTF with instructions significantly outperforms both PbRL and sentiment feedback, which do not have direct access to the ground truth feedback from the environment. It also significantly outperforms the setting where true trajectory-level feedback is received from the environment. In the case of RLHTF without instructions on how to construct the feedback, the performance is statistically indistinguishable from PbRL ($p = 0.5531$). Receiving the ground truth state-level feedback from the environment yields the best trajectories after four pieces of feedback, significantly outperforming all other settings, except for RLHTF with instructions where the difference is not statistically significant ($p = 0.1752$).

Table 1 depicts the performance of the final policy learned from 4 pieces of feedback. Most algorithms fail to recover the exact target trajectory, achieving 0% success rate. Notably, both our method *RLHTF with instructions* and ground-truth *state-level* feedback result in perfect trajectories in 25% of experiments. This demonstrates that natural language feedback, when guided with simple prompting instructions, achieve results comparable to the often unrealistic assumption of state-level supervision.

| | RLHTF without instructions | PbRL | Sentiment | True trajectory | True state |
|---|:---:|:---:|:---:|:---:|:---:|
| RLHTF with instr. | 0.0009 | 0.0024 | $3.4 \times 10^{-14}$ | $1.9 \times 10^{-8}$ | 0.1752 |
| RLHTF without instr. | - | 0.5531 | $2.1 \times 10^{-6}$ | 0.0391 | $6.4 \times 10^{-7}$ |
| PbRL | - | - | $2.7 \times 10^{-9}$ | 0.0021 | $4.8 \times 10^{-7}$ |
| Sentiment | - | - | - | $4.5 \times 10^{-5}$ | $2 \times 10^{-23}$ |
| True trajectory | - | - | - | - | $1.2 \times 10^{-16}$ |

Table 4: $p$-values for Welch's t-tests, showing that RLHTF with instructions and true state-level rewards significantly outperform other methods with 4 pieces of feedback.

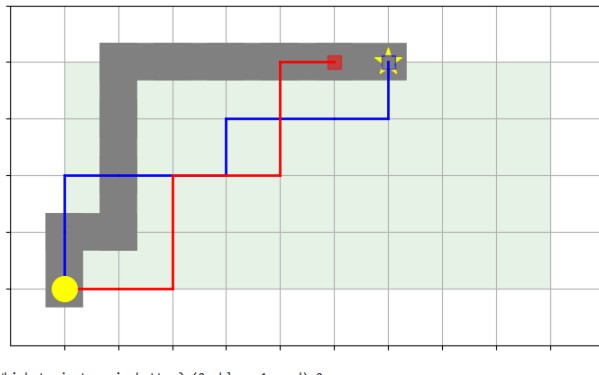

Which trajectory is better? (0: blue, 1: red) 0

Figure 9: Example of human interaction with PbRL in *Gridworld*.

**Guidelines for Evaluators**

For both RLHTF settings, we provided participants with the following guidelines:

> **Objective**: The purpose of this experiment is to provide instructional feedback to an artificial agent. The task for the agent is to navigate from a yellow circle to a yellow star along a gray pathway. The agent will attempt this task by following a trajectory marked in blue. Your role is to offer written feedback that assists in correcting the agent's current course.
>
> **Task Instructions**:
> 1. Observe the blue trajectory that the agent has taken.
> 2. Provide your guidance and feedback on the agent's performance (e.g., "Do not go below the sofa. The end was very good").
> 3. Repeat 4 times.

In the setting with instructions, participants also received the additional set of instructions outlined in Section 5.2. These instructions helped participants adapt their feedback more effectively, which in turn accelerated the learning of the reward model.

For the PbRL the participants were shown the example in Figure 9 and told:

> **Objective**: The purpose of this experiment is to provide feedback to an artificial agent. The task for the agent is to navigate from a yellow circle to a yellow star along a gray path. Two agents will attempt this task by following a trajectory marked in blue and red respectively. Your role is to select which of the two trajectories is better, so that the agent can learn the correct path.
>
> **Task Instructions**:
> 1. Observe the blue and red trajectories that the agents have taken.
> 2. Choose the best one (0: blue, 1: red).
> 3. Repeat 4 times.

Lastly, in the sentiment setting participants were told:

> **Objective**: The purpose of this experiment is to provide feedback to an artificial agent. The task for the agent is to navigate from a yellow circle to a yellow star along a gray path. The agent will attempt this task by following a trajectory marked in blue. Your role is to offer written feedback that assists in correcting the agent's current course.
>
> **Task Instructions**:
> 1. Observe the blue trajectory that the agent has taken.
> 2. Provide a sentiment rating (how good or how bad) for the trajectory, e.g.: "it is horrible" or "amazing".
> 3. Repeat 4 times.

### Additional Guidelines Effect on Consensus

To quantify annotator variability, we analyze the state-labels obtained across evaluators when exposed to the same 8 Gridworld test rooms with identical initializations. For each pair of participants, we compute the Jaccard similarity between their sets of state–label pairs, capturing the fraction of overlapping annotations relative to the total annotations made. In the setting without additional guidance, the average pairwise Jaccard similarity is 0.1028, indicating very high variability in how non-experts describe the same trajectories. When we provide additional guidelines to the evaluators, the average Jaccard similarity increases to 0.2241, so agreement between evaluators roughly doubles, although substantial variability remains. This analysis confirms that human feedback can be highly noisy and underscores that our method is explicitly evaluated under such challenging, non-expert conditions.

We further hypothesize that these guidelines also reduce ambiguity in the human feedback, thereby increasing output consistency across different LLM models. To test this hypothesis, we input the same prompts and collected feedback to 5 different models (sonar, gpt-4o, o1, o3-mini and gpt-3.5) and compared the output state-reward pairs. Figure 10 categorizes the generated state-reward pairs according to the number of models that agreed on them. We notice that by providing guidelines led to a marked shift toward consensus. The proportion of outputs with majority agreement (at least 3 models) rose from 48.5% to 73.3%. Specifically, complete agreement across all five models nearly doubled (10.3% to 20.4%), while inconsistent outputs generated by only a single model dropped significantly (30.1% to 13.5%).

Additionally, Figure 11 shows that the correctness of the output state-reward pair is highly correlated to its level of agreement between models. While 84% of the state-reward pairs generated by all the 5 models are correct, only 43% of the state-reward pairs generated by a single model are correct. This suggests that multimodal verification could help reduce hallucinations.

### Performance for Different LLMs

We assess the robustness of reward learning from human feedback by experimenting with alternative LLMss. Specifically, we reuse the human feedback collected during the RLHTF experiments with GPT-4o outlined in Section 5, and apply the same prompts to different base models. Figure 12a shows the average error evolution in the setting where human evaluators are not given additional instructions, while Figure 12b shows the performance when the feedback was given after observing additional instructions.

We observe a correlation between LLM capability and RLHTF performance. GPT-3.5 Turbo maintains the worst average error across interactions in both settings. Perplexity's sonar-reasoning model performs the best, showcasing its ability to interpret free form human feedback. Comparing final errors at interaction 4, Sonar-reasoning reduces the average error by 33% relative to GPT-3.5 in the no-instructions setting (from 6.7 to 4.5), and by 52% in the with-instructions setting (from 4.7 to 2.3). The superior performance of more capable models is also reflected in the Gridworld success rates summarized in Table 5, where more powerful LLM models achieve the target path 2 to 3 times more often than GPT 3.5 in the case with instructions.

The error reduction pattern across interactions follows a consistent trend: rapid improvement after receiving the first piece of text feedback, followed by more gradual improvements in subsequent interactions. This

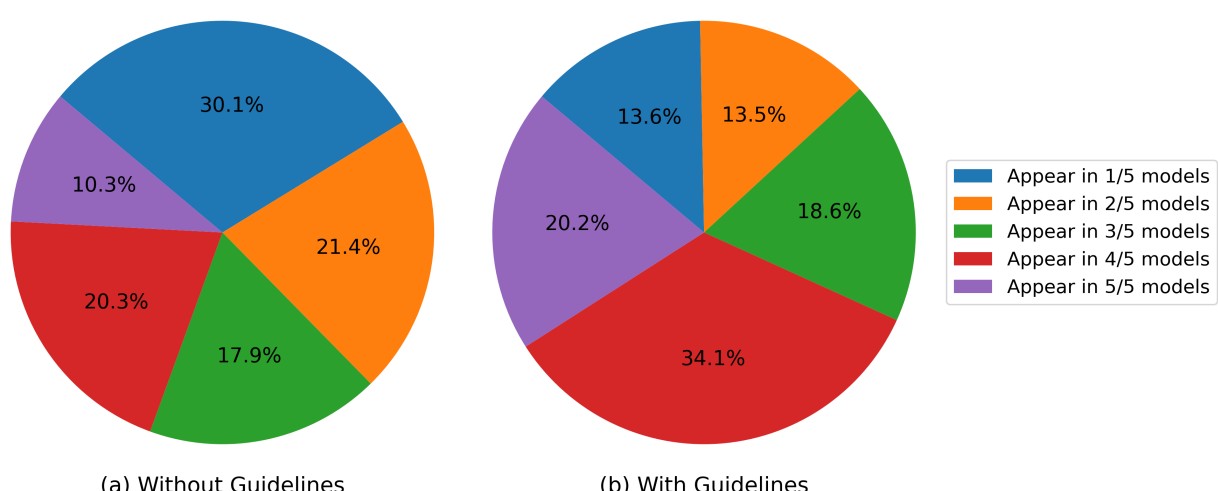

(a) Without Guidelines       (b) With Guidelines

Figure 10: Consensus of output state–reward pairs across five LLMs: sonar, gpt-4o, o1, o3-mini, gpt-3.5. Percentages reflect the share of all generated outputs, grouped by the number of models in agreement. We observe that providing guidelines to the evaluators increases the agreement between models.

| LLM model | With Instructions | Without Instructions |
|---|---|---|
| **gpt-3.5-turbo** | 9.8% | 2.2% |
| **gpt-4o** | 25.5% | 0.0% |
| **gpt-4.5** | 25.2% | 8.8% |
| **o1** | 27.5% | 4.4% |
| **o3-mini** | 25.5% | 0.0% |
| **sonar** | 15.7% | 2.2% |
| **sonar-reasoning** | 31.37% | 4.4% |

Table 5: Gridworld success rate of different LLM models with and without instructions.

pattern is maintained across all models, showcasing the robustness of RLHTF, though the rate of improvement varies, with newer models generally showing more substantial error reductions per interaction. These findings support our conjecture that RLHTF will likely continue to benefit from ongoing advancements in LLMs technology.

## B.2 Reacher Environment

Each trajectory in the Reacher environment spans $T = 50$ frames. The standard observation space contains information about the goal (target location) and the performance (distance to target). However, to prove that the LLM is capable of deducing the performance solely from the human feedback, we filter this information from the user prompt. The LLM obtains a filtered state that includes the fingertip's location, arm's joint angles, and angular velocities. The filtering and preprocessing of the observation are detailed in Figure 13.

We use the LLM *gpt-4o* to translate the human feedback to state-reward pairs. To conduct a fair comparison with PbRL, we reconstruct the filtered states output by the LLM back into the complete original observation

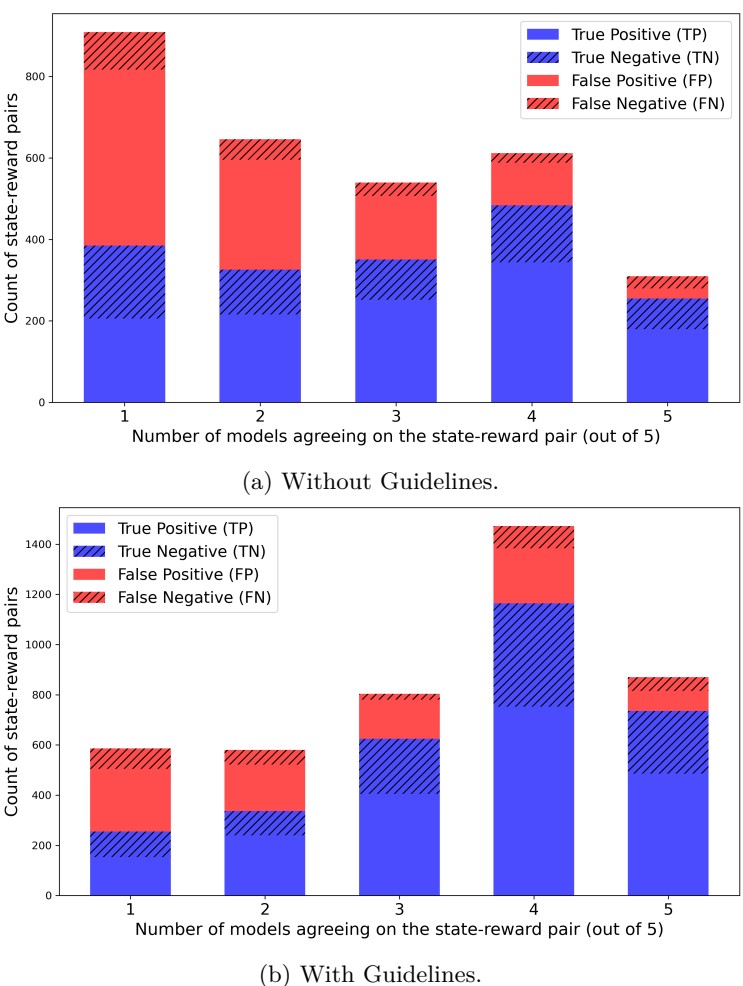

(a) Without Guidelines.

(b) With Guidelines.

Figure 11: Count of state-reward pairs for each level of model agreement, categorized by correctness (correct in blue, incorrect in red) and predicted labels (positive in solid, negative in hatched). A state-reward is correct if the reward accurately identifies whether the state is in the target path or not. The more models output a state-reward pair the higher the probability that it is correct.

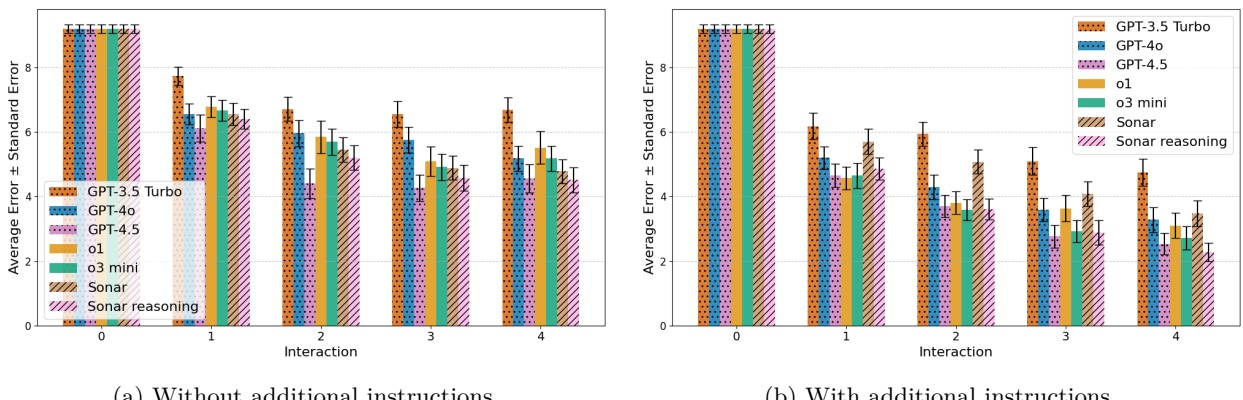

(a) Without additional instructions.

(b) With additional instructions.

Figure 12: Performance for different LLMs with feedback from RLHTF experiments

| Default Observation | |
| --- | --- |
| cosine of the angle of the first arm | 1.3184 |
| cosine of the angle of the second arm | 1.3184 |
| sine of the angle of the first arm | 0.9996 |
| sine of the angle of the second arm | 0.9995 |
| x-coordinate of the target | -0.0288 |
| y-coordinate of the target | -0.0307 |
| angular velocity of the first arm | -0.1320 |
| angular velocity of the second arm | 0.1214 |
| x-value of $position_{fingertip}$ - $position_{target}$ | 3.9565 |
| y-value of $position_{fingertip}$ - $position_{target}$ | 3.9607 |

| Filtered State | |
| --- | --- |
| Angles of arms in degrees | [5.8, 11.6] |
| Fingertip position in algebraic notation | x15 |
| Angular speed in rads/sec | [0.62, 3.73] |

Figure 13: Preprocessing of state observation in Reacher environment.

space. Using this full observations with their corresponding binary labels, we train a reward model in a supervised learning framework. The architecture of this reward model is a fully connected NN with one hidden layer consisting of 32 nodes, employing a ReLU activation function characterized by a leaky parameter $\alpha = 0.01$. In the case of PbRL, three such NNs are initialized at random to create an ensemble. Before asking the human for a preference, we sample random trajectories and then choose the pair whose preference has most uncertainty, specifically, the pair for which there is the most disagreement among the ensemble's predictive outcomes. During policy training, the agent observes the average reward from the ensemble.

We perform the experiments three times, each with a different target location, for every algorithm under consideration. The human feedback was provided by one of the authors of this paper. The shaded area in Figure 7 shows the standard error between the three experiments. Note that the plotted lines have been smoothed using a convolution operation with a window size of 100 episodes, which helps reduce noise and reveal clearer data trends.

### B.3   Rubik's Cube

Each trajectory consists of ten steps, during which the agent can perform one of 18 actions: rotating any layer clockwise or counterclockwise, or making no movement. The experiment begins with the Rubik's cube in its solved state, where each face consists of squares of a single color. By default, evaluators view the cube from the front-facing layer, but they can use keyboard keys to adjust the angle and observe other sides of the cube.

The Gym Rubik's cube environment observation is a 54-element array with values ranging from 0 to 5, representing the colors of the entire cube in numerical format. We slice the array to extract the 9 items corresponding to the front face and we reformat it into a 3x3 grid. We also map the numerical numbers to a letter representing the color; e.g., instead of '0' we write a 'W' to represent white. A list of ten $3 \times 3$ arrays, each representing the front face state at one timestep in the trajectory, is added to the user prompt, along with the text feedback. We don't include any landmarks in this environment.

Although the environment is discrete and countable, the vast number of possible states of a face makes tracking the reward distribution for each state impractical. To address this, we model both reward and policy functions with NNs. The policy function is parameterized by a two-layer neural network, where the first layer maps the observation space to a 128-dimensional latent representation, followed by a ReLU activation. The output layer maps this representation to the action space dimensions, applying a softmax activation to produce a probability distribution over actions. Similarly, the reward function is modeled as a feedforward neural network with a single hidden layer of size 8. The input is first transformed through a fully connected layer, followed by a ReLU activation. The output layer then produces a scalar reward value. The reward model is trained via supervised learning using the dataset generated by a pre-trained *gpt-4o* model from the human feedback. We train the agent with the REINFORCE algorithm (Williams, 1992).

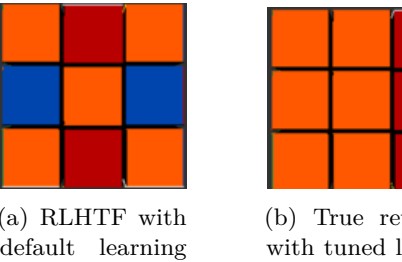

(a) RLHTF with default learning rate

(b) True reward with tuned learning rate

Figure 14: Obtained pattern in the front face when aiming to create an orange X (a) RLHTF achieves the desired pattern with 10 human feedback and default hyperparameters whereas (b) the true reward with tuned learning rate converges to a local maximum after 90,000 iterations of REINFORCE.

Figures 14 show the pattern the agent converges to when tasked with obtaining an orange 'X' on the front face of the cube. With the true environment reward, the agent converges to a local maximum, resulting in only three of the required squares being orange. In contrast, RLHTF successfully learns the desired pattern. To produce the results in Figures 14a and 8b, one of the authors acted as the evaluator and provided the following ten pieces of text feedback:

1. The first 4 states are very bad. The states at time7, 8 and 9 are all wrong. The goal is to have an orange X (orange squared in both diagonals) in the front side.

2. The first 7 moves are wrong. At time 8 if you add three orange squares (two on the left corners and one in the middle) it would be good. The goal is to have an orange X by using 5 orange squares in the diagonals.

3. Until state 3 and including state 3 it is all wrong. At state 4 we should have three more orange squares on the left corners and in the middle. States 5, 6,7 and 8 are very bad. State 9 with orange squares on the left corners and the middle would be good. The goal is to have 5 orange squares arranged in the shape of an X.

4. Until state 5 and including state 5 it is all wrong. At state 6 we should have three more orange squares on the left corners and in the middle. States 7 and 8 are very bad. State 9 with orange squares on the left corners and the middle would be good. The goal is to have 5 orange squares arranged in the shape of an X.

5. From 0 to 3 (including both) the states are wrong. The states at time 6 and at time 9 are perfect. The states 5, 7 and 8 are not completely correct as they need two extra orange squares on the right corners. The goal is to have 5 orange squares in the shape of an X.

6. From 0 to 3 (including both) the states are wrong. The states 6 and 10 are perfect. The states 7, 8 and 9 are not completely correct. The goal is to have 5 orange squares in the shape of an X.

7. The states 6 and 10 are perfect. The states 7, 8 and 9 are not completely correct. The goal is to have 5 orange squares in the shape of an X.

8. The states 6 and 10 are perfect. The states 7, 8 and 9 are not completely correct. The goal is to have 5 orange squares in the shape of an X.

9. The state 2 is very very bad. The states 6 and 10 are perfect. The states 7, 8 and 9 are not completely correct. The goal is to have 5 orange squares in the shape of an X.

10. The states 2 and 3 are very very bad. The states 6 and 10 are perfect. The states 7, 8 and 9 are not completely correct. The goal is to have 5 orange squares in the shape of an X. It could be an orange X with all the other squares in red or all the other squares in green.

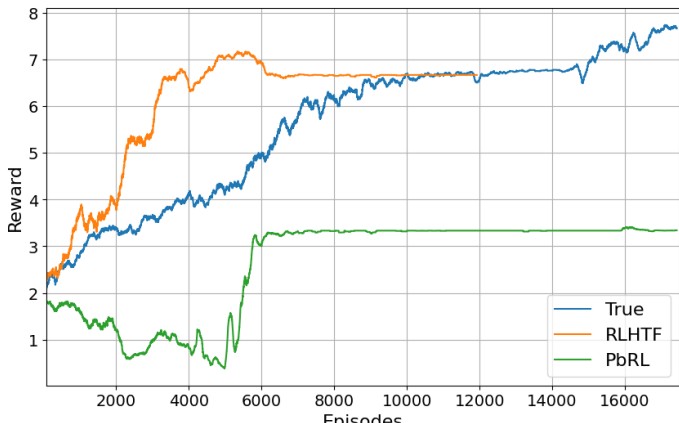

Figure 15: Reward evolution with episodes of the REINFORCE algorithm in the Rubik environment when aiming to construct an Italian flag. We compare the performance for RLHTF with 5 text feedback, PbRL with 5 trajectory comparisons, and true environment reward.

We performed a similar experiment aiming to recreate the Italian flag pattern (three vertical stripes: red, white and green) on the front side of the Rubik's cube. The results are shown in Figure 15. Remarkably, with only five pieces of human feedback, RLHTF nearly doubles the reward achieved by PbRL. However, RLHTF converges to a reward of 6.67, not to the maximum achievable score of 7.88. This discrepancy may arise because the performance appears 'good enough' to the human evaluator, in the sense that it already fulfills the task. In contrast, when having access to the true reward, the agent may continue learning until it discovers the optimal way to achieve the task. Nevertheless, this 'good enough' level is reached faster with just five pieces of human text feedback than when using the true environment reward at every REINFORCE step.

To better interpret performance trends, we smooth the rewards per episode in Figures 8b and 15 by doing a convolution with a window size of 30.

## C   Policy Learning

The reward model learns to measure how close the trajectory is to human intentions. Consequently, to find the optimal policy, the agent may directly query the reward model, instead of the human evaluators. This approach significantly reduces time, energy, and monetary costs during policy learning (Christiano et al., 2017). The interactions are described by the solid arrows in Figure 2.

The agents follow a greedy policy, they select actions that maximize the expected future rewards as in (1). In simple tabular settings, we use dynamic programming to find the optimal solution. In more complex or continuous environments, we model the policy with a fully connected NN, which takes the observed state as input and outputs an action. We train the network using the REINFORCE algorithm (Williams, 1992), where the reward signal is given by the reward model. Following standard practices from RLHF, we compute several REINFORCE epochs before querying the human evaluators and updating the reward model again.

We would like to highlight that our contribution lies in efficient reward modeling with fewer but richer human interactions. The agent learning phase, which takes most computational burden, follows standard RL algorithms, such as REINFORCE, whose computational complexities are well-established in prior literature. We run all our experiments on an Intel(R) Core(TM) i7-7800X CPU @ 3.50GHz processor, most learning was done in minutes and no experiment took longer than a couple hours.

An agent is trying to learn and follow a specific path in a {grid_height}x{grid_width} grid map.

Your job is to translate the feedback of the current trajectory into feedback types, locations in the map, and a label.

- If the feedback type is imperative, compute what locations in the map the instructions are referring to and label them as either 'good' (go to) or 'bad' (avoid).

- If the feedback type is evaluative, determine what locations in the map are being referred to by the feedback and whether the feedback is positive or negative.

CoT

- If the feedback type is descriptive, compute what new locations the agent should have visited, and label them as positive.

Use the getReward function to only return a JSON file with the specified shape enclosed in double quotes.

Output format

For example: if the user's input is {example_feedback}, then the output should be {example_output}.

Another example: if the user's input is {example_feedback2}, then the output should be {example_output2}

Few Shot

Figure 16: System prompt for Gridworld.

## D  Prompt Engineering

The system prompt design has a strong impact on the algorithm's performance. Next, we explain some of the design choices.

We define the coordinates in the grid using *chess style notation*, i.e., as a letter and number pair indicating the column and row respectively. This format ensures clear spatial referencing, even with older LLM versions like GPT 3.5, which often struggle with traditional Cartesian coordinates, e.g. $(2, 3)$, as it can be ambiguous whether the first number refers to the horizontal or vertical coordinate. With chess notation, we remove this ambiguity and improve the model's ability to interpret spatial information correctly.

Additionally, we follow Chain of Thought ($CoT$) prompting (Wei et al., 2022) to enhance reasoning when processing human feedback. Since feedback is inherently context-dependent, we introduce an intermediate classification step where the LLM categorizes the feedback into different types, such as goal description, imperative instruction or trajectory evaluation. This categorization helps structure the interpretation of feedback based on its intent.

In accordance with *few-shot prompting* (Kaplan et al., 2020), we provide demonstrations to steer the model to better performance. By exposing the model to relevant cases, we reduce ambiguity and improve performance.

Lastly, we use *function calling* to force ChatGPT's output to have a prespecified json format, which enables us to seamlessly extract the necessary information in downstream tasks. The desired output format for the Gridworld, Reacher and Rubik's cube environments are described in Figures 18, 21 and 24 respectively. While the full system prompts are shown in Figures 16, 17, 19, 20, 22, and 23.

```
grid_height = 5
grid_width = 10
max_col_letter = 'j'

example_feedback = '''
        {"feedback": "it should not go below the bed",
        "landmarks": {"clock": ["2f"], "bed": ["2c", "2d"]},
        "trajectory": ["1b", "1c", "1d", "1e", "1f", "1g", "2g", "3g", "4g", "4h"]}
        '''
example_output = {'locations': ['1c', '1d'], 'label': 'NEG', 'feedback_type': 'imperative'}

example_feedback2 = '''
        {"feedback": "the last couple steps are good",
        "landmarks": {"clock": ["2f"], "bed": ["2c", "2d"]},
        "trajectory": ["1b", "1c", "1d", "1e", "1f", "1g", "2g", "3g", "4g", "4h"]}
        '''
example_output2 = {'locations': ['4g', '4h'], 'label': 'POS', 'feedback_type': 'evaluative'}
```

Figure 17: Parameters for system prompt in Gridworld.

```
"name": "getReward",
"parameters": {
      "type": "object",
      "properties": {
        "locations": {
          "type": "array",
          "items": {
            "type": "string",
            "pattern": [1-{grid_height}][a-{max_col_letter}]
          },
          "description": ("Locations in the grid referring to feedback with row numbers and a lowercase letter "
                "for columns. The rows are numbered from bottom to top (1 is the lowest row, increasing "
                "as you move upward), and columns are labeled from left to right (a-j). For example, '1b' "
                "refers to the lowest row in the second column from the left.")
        },
        "label": {
          "type": "string",
          "enum": ["POS", "NEG"],
          "description": "The feedback's connotation, positive or negative."
        },
        "feedback_type": {
          "type": "string",
          "enum": ["imperative", "evaluative", "descriptive"],
          "description": """
          Imperative: Feedback includes instructions on what locations are good or should be avoided.
          Evaluative: Feedback is an assessment of the current trajectory.
          Descriptive: Feedback is about modifications for an improved trajectory.
          """
        }
      },
      "required": ["locations", "label"]
  }
}
```

Figure 18: Function calling to force output format in Gridworld.

You will return state-reward pairs for a two-joined robotic arm named 'Reacher-v4'.

The arm aims to reach targets with its end effector (fingertip), and you must assess what states are good or bad based on human observers' feedback.

As input, you will receive:

1. The natural language comment by a human observer who has seen the simulation.

2. The location of landmarks which are circles of different colors.

3. Trajectory of a simulation of the robot trying to reach a target. Each timestep is described by:

   a) The fingertip position - letter representing column (left 'a' to right 'z') and number representing row (down '0' to up '26')

   b) A 2 item list of angles in degrees corresponding to the first and second joint respectively.

   c) A value of 0 in the second joint means the arm is fully bent, while it is completely straight when it is -180 or 180.

   d) A 2 item list with the angular speed on the first and second joint respectively.

For each set of observer comments, use the provided trajectory and landmarks to determine successful and unsuccessful states.

Follow this steps

1. Classify each section (sentence or linked group of sentences) in the feedback text as:

a) Goal description: It describes where the target is, or where the fingertip should go (e.g.: You should go to the pink dot).

b) Trajectory feedback: It criticizes the simulation observed (e.g.: The first two steps are wrong).

c) Trajectory suggestion: It describes ways to improve upon states in the simulation observed (e.g.: Go a bit to the right of the state at time 23).

2. Generate state reward pairs depending on the feedback type

a) For Goal description: Provide a "reward": +1, "angular_speed": [0, 0] at the location of the described target position.

b) For Trajectory feedback:

2.1. Determine whether the feedback has a positive ("reward": +1) or a bad ("reward": -1) connotation.

2.2. Determine what state or states of the simulation it is referring to, and get the fingertip position, angles and angular speed of those locations

c) For Trajectory suggestion:

2.1. Determine what state or states of the simulation it is referring to, and get the fingertip position, angles and angular speed of those locations

2.2. Correct the states as suggested by the feedback and pair with a "reward": +1

CoT

Use the getReward function to only return a JSON file with the specified shape.

Output format

Figure 19: System prompt for Reacher environment Part 1.

Example inputs with expected outputs are provided below for guidance:

Example 1:

Input:

\{'feedback': 'The last half is very bad, it should go a bit higher than the blue dot',

 'landmarks': \{'yellow': 'b3', 'blue': 'l6', 'white': 'm7', 'orange': 'c23'\},

 'fingertip_position':  ['k15', 'k14', 'k14', 'k14', 'k13', 'l12'],

 'angle': [[30.7, -63.9], [33.2, -68.2], [35.7, -72.6], [38.3, -76.9], [40.8, -81.3], [43.2, -85.7]],

 'angular_speed': [[0.1, -0.06], [0.12, -0.12], [0.13, -0.17], [0.15, -0.23], [0.16, -0.29], [0.18, -0.34]] \}

Expected Output:

\{"referred_steps": [

  \{"fingertip_position":  'k14', "angle": [38.3, -76.9], "angular_speed": [0.15, -0.23], "reward": -1\},

  \{"fingertip_position":  'k13', "angle": [40.8, -81.3], "angular_speed": [0.16, -0.29], "reward": -1\},

  \{"fingertip_position":  'l12', "angle": [43.2, -85.7], "angular_speed": [0.18, -0.34], "reward": -1\},

  \{"fingertip_position":  'l7', "angular_speed": [0, 0], "reward": 1\},

  \{"fingertip_position":  'l8', "angular_speed": [0, 0], "reward": 1\}]\}

Example 2:

Input:

\{'feedback': 'The fifth step but slower is good. Stop at the last point. The goal is to go to the white point.',

 'landmarks': \{'yellow': 'u8', 'purple': 'k12', 'white': 'w16'\},

 'fingertip_position':  ['k15', 'k14', 'k14', 'k14', 'k13', 'l12'],

 'angle': [[12.2, 34.2], [11.3, 39.6], [10.0, 45.2], [8.2, 50.9], [5.9, 56.6], [3.1, 62.3],],

 'angular_speed': [[0.1, -0.06], [0.12, -0.12], [0.13, -0.17], [0.15, -0.23], [0.16, -0.29], [0.18, -0.34]] \}

Expected Output:

\{"referred_steps": [

  \{"fingertip_position":  'k13', "angle": [5.9, 56.6], "angular_speed": [0.08, -0.15], "reward": 1\},

  \{"fingertip_position":  'l12', "angle": [3.1, 62.3], "angular_speed": [0, 0], "reward": 1\},

  \{"fingertip_position":  'w16', "angle": [13.0, 10.1], "angular_speed": [0, 0], "reward": 1\},]\}

Few Shot

Figure 20: System prompt for Reacher environment Part 2.

```
_pattern = '^26|[1-3]?[0-9]{1,2}[a-z](?:[a-k])?$'
FUNCTION_STRUCTURE = {
  "name": "getReward",
  "parameters": {
    "type": "object",
    "properties": {
      "referred_steps": {
        "type": "array",
        "items": {
          "type": "object",
          "properties": {
            "fingertip_position": {
              "type": "string",
              "pattern": _pattern,
              "description": ("Location of the fingertip with row numbers and a lowercase letter "
                      "for columns. The rows are numbered from bottom to top (1 is the lowest row, increasing "
                      "as you move upward), and columns are labeled from left to right (a, b, ..., z). "
                      "For example, '1b' refers to the lowest row in the second column from the left.")
            },
            "angle": {
              "type": "array",
              "items": {
                "type": "number",
                "format": "float",
                "minimum": -180,
                "maximum": 180,
                "description": "Angle in degrees for the first and second joint of the arm."
              },
              "minItems": 2,
              "maxItems": 2,
              "description": "A 2-item list of angles in degrees corresponding to the first and second joint
respectively."
            },
            "angular_velocity": {
              "type": "array",
              "items": {
                "type": "number"
              },
              "minItems": 2,
              "maxItems": 2,
              "description": ("Vector of two elements corresponding to the angular velocity of the first and second
arm respectively."
                      "If feedback refers to a location with positive reward, but without specifying the speed, set the
angular speed to [0, 0].")
            },
            "reward": {
              "type": "integer",
              "enum": [-1, 1],
              "description": "The reward value, which can be +1 for good performance or -1 for bad performance."
            }
          },
          "required": ["fingertip_position", "reward"],
          "minProperties": 2,
          "description": ("Dictionary containing information about the state and its reward. "
                  "Must have the 'reward' and 'fingertip_position' keys and optionally other keys: "
                  "'angle', or 'angular_velocity'.")
        },
        "description": "List of states described by the feedback along with their reward implied by the feedback."
      }
    },
    "required": ["referred_steps"]
  }
}
```

Figure 21: Function calling to force output format in Reacher environment.

You will assess the state-reward pairs of the front face of a Rubik's cube based on human observer feedback with the goal of achieving a specific pattern.

**Objective:** Identify successful and unsuccessful states of the Rubik's front face cube based on observer comments after viewing a 10-move simulation.

**Input Format:**

1. **Observer Comments**: Natural language comments by a human observer who has seen the simulation.

2. **Rubik's Cube Trajectory**: States of the Rubik's cube for 11 timesteps:

    • You will receive 11 states (initial state + 10 subsequent states).

    • Cube State: Defined by a 3x3 matrix for the front face of the cube, each cell representing a color (e.g., [[R, G, B], [W, Y, O], [B, R, G]]).

**Processing Steps:**

For each sentence or comment in the human feedback:

1. **Classify Human Feedback**:

    • Goal Description: Describes the target pattern for the cube (e.g., "You should have all red squares on the middle column.").

    • Trajectory Feedback: Criticizes the observed simulation (e.g., "The first two steps are wrong, but the setup at time 5 was good").

    • State Suggestion: Suggests corrections to the cube's state (e.g., "At time 3 you should have another red square on the top right corner").

2. **Generate State-Reward Pair**:

    • Goal Description:
        • reward: +1
        • state: The target state as described by the comment (a 3x3 matrix representing desired colors).

    • Trajectory Feedback:
        • Determine the connotation of the feedback (positive: reward: +1, negative: reward: -1).
        • Identify the index of the specific state(s) the feedback refers to, and return the input state(s) corresponding to such index.

    • State Suggestion:
        • Identify the index of the state referenced.
        • Modify the state as suggested.
        • reward: +1.

**Return the Results:**

    • Check your results

    • Use the getReward function to only return a JSON file with the specified shape.

CoT

Output format

Figure 22: System prompt for Rubik's cube environment Part 1.

Example inputs with expected outputs are provided below for guidance:

Example 1:

Input:

\{'feedback': 'The top row should be all white and there should be a blue in the lower right corner.',

'state0': [['R', 'B', 'G'], ['G', 'Y', 'R'], ['O', 'B', 'W']],

'state1': [['B', 'W', 'O'], ['G', 'Y', 'R'], ['O', 'B', 'W']] \}

Expected Output:

\{"state":  [['W', 'W', 'W'], ['G', 'Y', 'R'], ['O', 'B', 'B']]},

"reward": +1\}

Example 2:

Input:

\{'feedback': 'The end is bad. The last state with a yellow on the bottom left, and another yellow on the top of the middle column of the front side would be good.',

'state0': [['B', 'W', 'O'], ['G', 'R', 'Y'], ['O', 'B', 'R']],

'state1': [['B', 'W', 'R'], ['G', 'R', 'W'], ['O', 'B', 'G']],

'state2': [['B', 'G', 'R'], ['G', 'R', 'W'], ['O', 'B', 'G']] \}

Expected Output:

\{"state":  [['B', 'G', 'R'], ['G', 'R', 'W'], ['O', 'B', 'G']] ,

"reward": -1]\},

\{"state": [['B', 'Y', 'R'], ['G', 'Y', 'W'], ['Y', 'B', 'G']],

"reward": +1]\}

Few
Shot

Figure 23: System prompt for Rubik's cube environment Part 2.

```
FUNCTION_STRUCTURE = {
 "name": "getReward",
 "parameters": {
  "type": "object",
  "properties": {
   "states": {
    "type": "array",
    "items": {
     "type": "object",
     "properties": {
      "state": {
       "type": "array",
       "description": "A 3x3 grid representing the state of the board. Each subarray corresponds to a row from left to
                       right, with the first subarray representing the top row, the second representing the middle
                       row, and the third representing the bottom row.",
             "items": {
        "type": "array",
        "items": {
         "type": "string",
          "enum": ["W", "G", "O", "B", "R", "Y"]
         },
         "minItems": 3,
         "maxItems": 3
        },
        "minItems": 3,
        "maxItems": 3
       },
       "reward": {
        "type": "integer",
        "enum": [1, -1]
       }
      },
      "required": ["state", "reward"]
     }
    }
   },
   "required": ["states"]
  }
}
```

Figure 24: Function calling to force output format in Rubik's cube environment.

