# OpenReview forum: "From Words To Rewards: Leveraging Natural Language For Reinforcement Learning"
_TMLR — Accepted by TMLR_

### Review · Reviewer_R64J · 2025-11-07

**Summary Of Contributions:**

This paper proposes  Reinforcement Learning from Human Text Feedback (RLHTF). It allows the LLM to recieve the human's feedback along with the reward to improve the performance. The proposed method contains two phase: Reward learning (described in Section 4) and policy learning. In experiments, the proposed method achieves better performance.

**Audience:**

Yes

**Audience Explanation:**

As RLHF is a hot topic, many audience would be interested in knowing this result.

**Broader Impact Concerns:**

No any broader impact concerns.

**Claims And Evidence:**

Yes

**Claims Explanation:**

All four contributions are clearly supported in this paper. For its constribution 1 and 2, the proposed RLHTF framework indeed ultilizes the human feedback to enhance the reward model in the RL fine-tuning. For its contribution 3 and 4, the empirical experiments support that the proposed method achieves better performance. As a result, these claimed contributions are supported by these evidences.

**Requested Changes:**

It is hard to connect Section 2 Problem Setting with existing LLM fine-tuning route. It would be better if the author could clearly clarify how each part is defined in RLHF. For example, what are the state, action, transition, and reward defined in RLHF and the proposed RLHTF setting. Currently, Section 2 is just a general description of RL, which is not very useful to understand the problem setting.

On page 3: "Analogously to other RLHF algorithms,". I am not sure if other RLHF algorithms will require to train a reward model. In many math reasoning tasks, the reward model is just a text parser to extract the correct answer from the model's response. It doesn't really need to train an additional model. I may suggest the author to further clarify here to explicitly state the detailed setting.

---

> ### Author Response · Authors · 2025-12-02
> **Response to Reviewer R64J**
>
> We are grateful for the reviewer's positive assessment of our work and for recognizing that our “contributions are clearly supported” and observing that the TMLR “audience would be interested in knowing this result”. The suggested clarifications regarding the problem setting are valuable, and we have revised Section 3 Problem Setting accordingly.
> 1. In the revised version, we make the mapping between our MDP formulation and the RLHF setting explicit. The MDP\R is the mathematical framework of the problem, and RLHF is a “solution approach” that attempts to learn both the unknown reward model and the optimal policy. The revised paper clarifies what the state, action, transition and reward are by providing some examples. As the reviewer suggests, many people are aware of RLHF in the context of LLM fine-tuning. We now better frame our setting by explicitly mentioning the components’ meaning in this popular context. The updated paragraph reads as follows:
>
>     *Formally, an episodic Markov Decision Process without reward function (MDP$\backslash$R) is a tuple $\mathcal{M} := (\mathcal{S}, \mathcal{A}, \mathbb{P}, T)$. $\mathcal{S}$ is the state space where each state $\mathbf{s}\in\mathcal{S}$ captures the environment configuration. For example, the position of an agent in a grid, the position of a car and all surrounding objects in a self-driving task or the prompt and partial answer in LLM fine-tuning. $\mathcal{A}$ is the set of actions that the agent can perform in the environment; for example, an action could be a step to the right on the grid, accelerating a car or generating a specific next token in LLM fine-tuning. $\mathbb{P}: \mathcal{S} \times \mathcal{A} \rightarrow \Delta (\mathcal{S})$ captures the transition probabilities, mapping state-action pairs to a probability distribution of the next state over $\mathcal{S}$, where $\Delta (\mathcal{S})$ denotes the probability simplex over $\mathcal{S}$, and $T$ is the time horizon. $\mathbb{P}$ captures the environment dynamics, such as the distribution of a car's position after acceleration or how text evolves after token generation. The reward function, which maps state-action pairs to a reward $r: \mathcal{S} \times \mathcal{A} \rightarrow \mathbb{R}$, is unknown to the agent. Instead, the agent learns a reward model $\widehat{r}: \mathcal{S} \rightarrow \mathbb{R}$ based on human feedback.  The unknown reward function reflects how well state-action pairs align with the human's implicit preferences or goals.*
>
>
> 2. The reviewer is correct to point out that not all RL pipelines require training a separate reward model from human feedback, for example in verifiable tasks such as math problems where a parser can directly compute a reward from the output. However, we target solely settings where such a verifiable reward is not available or is difficult to specify, and where a reward model must instead be learned from human feedback. We now clarify the statement in page 3 within the broader landscape accordingly and say “*Analogously to the classical RLHF framework (Christiano et al., 2017).*”

---

### Review · Reviewer_rnic · 2025-11-09

**Summary Of Contributions:**

This paper proposes RLHTF (Reinforcement Learning from Human Text Feedback): using LLM to convert natural language feedback from humans into state labels to train a reward model. Specifically, the overall pipeline is divided into three steps: 1) Context encoding: packaging the execution trajectory, corresponding human feedback, and some information into a formatted prompt; 2) Translation: feeding the above prompt into the LLM to obtain data with state labels; 3) Training the reward model: training the reward model using the data obtained in Step 2. The above process can accelerate convergence and improve learning efficiency.

**Audience:**

Yes

**Audience Explanation:**

RLHF (and human-computer alignment) is an important topic in itself, and readers will likely find it interesting. At the same time, the RLHTF proposed in this paper is a concise and clear method that helps achieve more efficient alignment, making it attractive to the community.

**Broader Impact Concerns:**

1. The article mentions recruiting volunteers for the experiment, but the authors have clearly stated that minimal risk research also provides compensation.
2. Feedback from small groups of participants may introduce some biases (e.g.,  cultural/gender/language).

**Claims And Evidence:**

Yes

**Claims Explanation:**

1. Clear method: Clearly explain the core of the method and the overall pipeline: Context encoding -> Translation -> Training the reward model
2. Enough experiments: The paper compares the RLHTF with multiple baselines (true trajectory/true state, sentiment score, PbRL) in various environments, and reports metrics such as success rate, error/reward curve, and significance, and provides analysis and conclusions.
3. Detailed analysis: Analysis of performance, efficiency, and details.

**Requested Changes:**

1. The proposed method relies on prompts, which are specifically designed and have limited generalization ability. This limitation is discussed in the authors' future work; further related preliminary experiments or a more in-depth discussion could be considered.
2. The proposed method relies on the translation capabilities of LLM (step 2), a process prone to hallucinations. The authors currently mitigate this by providing guidance to evaluators, but this issue still requires further exploration. This approach also relies on human behavior. The authors could consider more mechanized approaches to alleviate this problem, such as back-translation consistency checks and multi-model consistency verification.
3. Consider providing an ethical statement explaining the potential risk according to the 'Broader Impact Concerns'.

---

> ### Author Response · Authors · 2025-11-26
> **Response to Reviewer rnic: Prompt Generalizability & Multi-Model Consistency**
>
> We thank the reviewer for providing insightful and constructive feedback. We are grateful that the reviewer found our approach to be a “concise and clear method that helps achieve more efficient alignment, making it attractive to the community.” Next, we address the requested changes:
>
> 1. We agree with the reviewer that an in-depth discussion of the **prompt generalizability** is valuable. *We extended its discussion in the future work section* as follows:
>
>     First, our prompts must be adapted to each environment, similar to how traditional RL requires handcrafting reward functions. Although we demonstrate some degree of generalizability across three distinct environments, fully task-agnostic prompting remains an open challenge. The system prompts require manual design to ground the LLM in the specific context of each environment. In our experiments, we partially reduce this burden by using off-the-shelf object detectors (i.e., YOLO) to automatically detect and name landmarks, which are then inserted into the prompt. However, the core system prompt still requires manual design. To further automate the system prompt generation, future work could leverage metadata (e.g., simulator documentation or object annotations) to automatically generate descriptions or relevant items, (e.g. the description of state elements). Alternatively, future work could explore automatic prompt optimization techniques  (Ramnath et al., 2025), that use evolutionary strategies to discover effective prompts without human intervention.
>
> 2. We thank the reviewer for suggesting that we examine **LLM output consistency** and whether we can utilize multi-model consistency verification to mitigate hallucinations. Next we describe the additional experiments and analysis we ran to address this point.
>
>     Our paper shows how giving guidelines on how to provide feedback increases the correctness of the state-reward pairs output by the LLM in the Gridworld experiments. We further hypothesize that these guidelines also reduce ambiguity in the human feedback, thereby increasing output consistency across different LLM models. To test this hypothesis, we input the same prompts and collected feedback from 5 different models (sonar, gpt-4o, o1, o3-mini and gpt-3.5) and compared the output state-reward pairs. Figure 10 (in the revised pdf) categorizes the generated state-reward pairs according to the number of models that agreed on them. We notice that *providing guidelines led to a marked shift toward consensus.* The proportion of outputs with majority agreement (at least 3 models) rose from 48.5% to 73.3%. Specifically, complete agreement across all five models nearly doubled (10.3% to 20.4%), while inconsistent outputs generated by only a single model dropped significantly (30.1% to 13.5%).
>
>     Additionally, Figure 11 shows that the *correctness of the output state-reward pair is highly correlated to its level of agreement between models.* While 84% of the state-reward pairs generated by all the 5 models are correct, only 43% of the state-reward pairs generated by a single model are correct. This validates Reviewer rnic suggestion that multimodal verification could help reduce hallucinations. Future work could deploy an ensemble method, where each piece of human feedback is interpreted by several models, and only outputs with majority agreement are retained. Our analysis suggests that keeping only the state-reward pairs on which the majority of models agree will filter many of the incorrect interpretations. We have added this idea to the future work section, and the corresponding figures and analysis to the appendix.

---

> > ### Author Response · Authors · 2025-11-26
> > **Response to Reviewer rnic: Broader Impact Statement**
> >
> > 3. We thank the reviewer for raising this ethical point. We now included the following **broader impact statement** at the start of the appendix:
> >
> > **Human Subject Research**
> >
> > Our work involves a study with human participants. The study was categorized as minimal risk research qualified for exemption status under 45 CFR 46 104d.2 by the Institutional Review Board (IRB). All 26 participants were adults, provided informed consent before participation, were able to withdraw at any time, and were equally compensated $10 for approximately 30 minutes of their time (no volunteer work was used). No sensitive information nor personally identifying information beyond coarse demographic data was collected, stored or shared.
> >
> > The ultimate goal of our work is to reduce the need for complex reward engineering by enabling alignment by non-experts from their feedback in the form of natural language. This new manner of alignment may contribute to the democratization of RL training. However, as with all alignment technologies, there is a dual-use risk where such methods could be employed to align agents with malicious instructions. Malicious actors could exploit natural language feedback interfaces to inject misleading rewards, causing RL agents to learn harmful behaviors. All our experiments were done in simulated environments with no deployment to real‑world systems or high‑stakes decision domains. Any future applications in safety‑critical or socially sensitive settings would require additional domain‑specific oversight and safeguards.
> >
> > The proposed algorithm seeks to align RL agents with fewer human interactions. However, employing fewer evaluators for alignment may exacerbate bias. This reinforces the importance of hiring diverse evaluators across cultural, gender and language backgrounds.
> >
> > **Reliance on LLMs**
> >
> > This work uses LLMs as translation functions to map human feedback onto state-level rewards. However, LLMs are prone to hallucinations or biased interpretations, especially when presented with ambiguous feedback. We attempt to mitigate these risks by using few-shot prompting, environment-specific prompts and providing evaluators with guidelines, however, misinterpretations still persist. Moreover, the performance of our approach depends on proprietary LLM models, which may limit transparency and reproducibility. To partially mitigate this, we release our prompts and code, and report results across multiple LLMs.

---

> > > ### Comment · Reviewer_rnic · 2025-11-27
> > > **Response to Authors' Rebuttal**
> > >
> > > Thank you for the detailed response. I am pleased to see the additional multi-model consistency experiments. Most of my concerns were addressed. Thank you for the rebuttal and careful revisions.

---

### Review · Reviewer_bg8B · 2025-11-18

**Summary Of Contributions:**

The paper proposes RLHTF, a framework where humans provide free-form natural language (NL) feedback on trajectories, and a large language model translates this text, together with simple “landmarks” and trajectory information, into state-level labels used to train a reward model. The authors study the effect of feedback granularity in a small Gridworld with a user study, and present additional proof-of-concept experiments on a low-dimensional Reacher task and a reduced Rubik’s Cube “front-face pattern” task, showing that RLHTF can outperform sentiment-based and preference-based baselines with relatively few feedback interactions.

**Key strengths**:
1. Timely and interesting direction: using rich natural language as a higher-bandwidth reward signal.

2. Clear high-level pipeline (context encoding → LLM translation → reward update).

3. Some empirical evidence (user study, multiple toy domains) that structured NL feedback plus an LLM can be more effective per interaction than PbRL/sentiment.

**Key weaknesses**:

1. All environments are very simplistic; there is no convincing demonstration on a realistically challenging RL task.

2. Baselines are narrow (mainly PbRL and sentiment); standard sparse-reward RL or GCRL baselines, which likely perform well here, are missing.

3. Critical methodological components (landmark extraction, LLM translation function, reward model update) are not accurately described in the main texts and heavily rely on environment-specific prompt/landmark engineering.

4. The writing repeatedly overstates what the experiments show, claiming to “prove” robustness in continuous and structurally complex environments, that RLHTF can “rival access to privileged ground-truth rewards,” and that it “overcomes reward modeling challenges” and will “continue to benefit from future advances in LLM capabilities.” These claims are based on a single small Gridworld, a simple 2-DoF Reacher task, and a heavily simplified Rubik front-face toy, all with carefully chosen baselines, which does not justify such broad, forward-looking statements.

**Audience:**

No

**Audience Explanation:**

While the general direction is undoubtedly interesting and relevant to TMLR, this particular submission does not, in my view, produce findings that are useful or general enough for the journal’s audience. The experiments are confined to very simplistic, highly engineered settings, and the methodological details are too vague to make the approach directly reusable. As a result, it is hard to distill any clear, general statement or actionable insight that a TMLR reader could take away and apply beyond the specific toy setups considered here. The work feels more like an early research than a mature contribution that would meaningfully inform or influence the broader community.

**Claims And Evidence:**

No

**Claims Explanation:**

The central claims are not convincingly supported by the current evidence, for several reasons:

1. **Toy-scale experiments vs broad claims**. The strongest claims (e.g., that RLHTF can “rival access to privileged ground-truth rewards,” “prove robustness” in continuous and structurally complex environments) are based only on a tiny tabular Gridworld, a very simple 2-DoF Reacher, and a heavily simplified Rubik’s “front-face pattern” toy. These settings are far too narrow and easy to justify such general statements about robustness, scalability, or practical impact.

2. **Missing strong baselines**. The method is mainly compared to preference-based RL and sentiment-based rewards. There are no comparisons against standard RL or GCRL methods with simple sparse rewards, which could almost certainly solve these environments. As a result, the claimed advantages over “baseline methods” may simply reflect the choice of weak baselines, not a genuine advantage of the approach.

3. **Under-specified methodology**. Critical parts of the pipeline are only described at a high level. Landmark extraction, the LLM “translation function” from free text to state-level labels, and the reward-model update procedure are all presented as black boxes with environment-specific prompt engineering and minimal algorithmic detail. This makes it hard to assess reproducibility or to understand when and why the method should work.

4. **Overinterpretation of fragile results**. Several headline claims rely on very specific, fragile phenomena (e.g., slightly better early error reduction than ground-truth feedback in one Gridworld protocol, or escaping a local optimum in one particular reward shaping for the Rubik toy). These are then generalized into broad narratives about overcoming reward-design difficulties and benefiting indefinitely from future LLM advances, which goes well beyond what the data actually support.

Taken together, the experiments provide some initial, suggestive evidence that natural-language feedback plus an LLM can be useful as a reward signal in simple settings, but they do not provide accurate, convincing, and clearly argued support for the stronger claims made in the paper.

**Requested Changes:**

1. **Stronger, non-toy experimental setting**.
Include at least one genuinely challenging RL task (e.g., high-dimensional / vision-based locomotion or navigation), make it the primary case study, and relegate most toy examples (Gridworld, simple Reacher/Rubik) to a supporting role.

2. **Add appropriate, strong baselines.**
Compare against standard RL / GCRL methods with reasonable sparse or shaped rewards (and tune them fairly), not just PbRL and sentiment. This is necessary to show RLHTF is useful beyond weak or contrived baselines.

3. **Fully specify the methodology.**
Provide enough detail to reproduce the method in the main text: how landmarks are obtained and encoded, the exact LLM prompting and output format for the translation function, and the full reward-model update procedure (architectures, losses, optimization, data aggregation).

4. **Analyze robustness and human-effort trade-offs.**
Include experiments with non-expert / noisy feedback and quantify the cost of feedback (number of interactions, words, or time) versus performance, including more realistic PbRL regimes.

5. **Align claims and writing with evidence.**
Remove or substantially soften overstatements (e.g., “prove robustness,” “rival ground-truth rewards,” “overcome reward modeling challenges”) and clearly present the work as an initial proof-of-concept with explicit limitations.

---

> ### Author Response · Authors · 2025-11-26
> **Response to Reviewer bg8B: Experimental Environments and Baselines**
>
> We thank the reviewer for the substantial feedback and the opportunity to clarify our contributions. We appreciate that the reviewer noted that our work is in a “timely and interesting direction”. Below, we address each of the reviewer’s points:
> 1. We acknowledge the reviewer’s concern about the **simplicity of the environments in our experiments**. However, our experimental design deliberately prioritizes *interpretability* in order to *isolate and validate our core methodological contribution*: showing that the proposed in-context learning method can successfully translate natural-language into an accurate training dataset of state-reward pairs for RLHF, a paradigm not previously explored in the literature. Reviewer rnic explicitly notes that we provide "enough experiments" with "detailed analysis" to support this capability. The three RL environments we study, spanning discrete and continuous control, offer interpretable states, rewards, and failure cases, which is crucial for auditing how textual specifications translate into supervision and for establishing the feasibility of this new RLHF paradigm. A fully general, *high-dimensional vision-based instantiation is beyond the scope of this work*. While scaling to such tasks is valuable future work it would introduce additional confounding factors that make it harder to diagnose whether limitations arise from the reward generation mechanism we propose or from domain complexity. Instead, our work shows a new paradigm for RLHF not explored before which as Reviewer R64j remarks “many audience would be interested in knowing,” and we believe it opens a new research direction upon which future work can build toward more challenging domains.
>
> 2. We agree with the reviewer that comparing to relevant **baselines** is key to show the advantage of our approach. However, standard RL and Goal Condition RL (*GCRL*) baselines are *not appropriate because they assume access to a reward function*, which is precisely what our method aims to learn. Our paradigm addresses situations where the reward function is not available or difficult to specify, e.g., when a user requests “a ‘cute’ pattern on the Rubik cube”. To emulate GCRL, where “the reward function is known and usually defined as a binary bonus of reaching the goal,” (Liu 2022) we could ask evaluators whether each trajectory achieves the goal. However, this binary feedback would create extremely sparse and minimally informative signals. Within the limited interaction budget (4 to 15 interactions used in our experiments), no trajectory reaches the goal, meaning almost exclusively zero reward would be observed, yielding a poor performance and not representing a meaningful baseline. We have updated Section 5 to clarify our baseline choices and explicitly explain why approaches that require access to a reward function, including GCRL, are not aligned with the setting we study.
>
>     More importantly, we compare our algorithm against substantially stronger baselines than those suggested by the reviewer. In particular, *we compare* the performance of the reward model extracted from text feedback *against the true trajectory-level reward and the true state-level reward, representing the strongest possible comparison* since these oracles have perfect knowledge of the true reward. In our experiments, RLHTF surpasses true trajectory-level feedback, directly addressing the concern that simple sparse-reward methods would trivially outperform our approach. We will clarify this in the revision to emphasize that our conclusions are validated against standard RLHF approaches (PbRL and sentiment) and true rewards, not weak baselines.

---

> > ### Author Response · Authors · 2025-11-26
> > **Response to Reviewer bg8B: Methodological Details**
> >
> > 3. We thank the reviewer for noticing we provide a “clear high-level pipeline.” Due to space constraints we are not able to provide all the **methodological details** in the main text. However, we will ensure that the main text points to the details to reproduce the methods present in the appendix. Here we provide the methodological implementation details that the reviewer asks about and state where to find them in our paper:
> >     - **Landmark extraction**: As detailed in Appendix B.1., the landmarks are detected with a YOLOv8 model trained on the Microsoft COCO dataset. [This jupyter notebook](https://anonymous.4open.science/r/WordsToRewards-2846/GridWorld/notebooks/task_generation.ipynb) contains the code to execute this landmark extraction.YOLOv8 outputs the landmark names and the pixels where they are detected, these pixels are translated into locations in the grids with chess style notation as further detailed in Appendix D.
> >     - **LLM translation function**: The LLM receives an input, a user prompt and a system prompt. The user prompt is detailed in Section 4.1. and it contains the text feedback provided by the user, the trajectory presented to the user, and the extracted environment landmarks. The system prompt is environment specific, as it describes the environment and the meaning of items in the user prompt. As Appendix D describes in detail, we use chain of thought and few-shot prompting to improve the performance. All system prompts appear verbatim in Appendix D. The output of the LLMs is a json file, with the specific format enforced by the function calling. Namely, as Figure 16 shows, for the Gridworld experiments the LLMs output a json object containing  an array of locations, a binary label (either ‘POS’ or ‘NEG’), and the feedback category specified during chain of thought. Similarly, as Figure 19 shows, the json object generated in the Reacher environment contains a dictionary where each element includes a fingertip position, angles of the joints of the arm, angular velocities, and a binary reward (either -1 or 1). Lastly, the function calling in the Rubik’s cube environment is shown in Figure 22. The json object contains pairs of reward and states, where the reward is represented by a 3x3 grid of colors, and the reward is binary (either -1 or 1).
> >     - **Reward model update**:  The discrete Gridworld environment only has 50 states, so as described in Section 5, we track “the reward probability distribution for every state, which we model as a beta distribution. In all scenarios, we initialize the reward distributions as beta(0.5, 0.5) to introduce a bias towards binary rewards: 0 (negative) or 1 (positive). For analytical tractability, we model the distribution of the observed rewards with the conjugate prior, i.e., as Bernoulli distributions.” Therefore, the posterior reward distribution, updated in a Bayesian manner, remains a beta distribution. For the more complex environments, we train a reward model in a supervised learning framework from the training dataset that we extend as more text feedback is translated. The exact details for the reward model updates are in Appendix B. In the Reacher environments, the architecture of the “reward model is a fully connected neural network (NN) with one hidden layer consisting of 32 nodes, employing a ReLU activation function characterized by a leaky parameter alpha = 0.01. “ For the Rubik’s cube experiments, the reward function is modeled as “a feedforward neural network with a single hidden layer of size 8. The input is first transformed through a fully connected layer, followed by a ReLU activation. The output layer then produces a scalar reward value. The reward model is trained via supervised learning using the dataset generated by the pre-trained gpt-4o model from the human feedback.”
> >
> >     We would like to highlight that *all exact prompts and code are publicly available*, ensuring the reproducibility of our method.

---

> > > ### Author Response · Authors · 2025-11-26
> > > **Response to Reviewer bg8B: Robustness and Claims**
> > >
> > > 4. We appreciate the reviewer highlighting the importance of analyzing **robustness with “non-expert / noisy feedback.”**  Our work already moves in this direction. Unlike most work in this area, which relies on synthetic or expert‑crafted feedback, we conduct experiments with *real, non‑expert participants* who provide free‑form language feedback *in the Gridworld task*. This setup explicitly exposes our method to annotator noise and variability, and represents a step towards real-world applicability.
> > >
> > >     To quantify annotator variability, we analyze the state-labels obtained across evaluators when exposed to the same 8 Gridworld test rooms with identical initializations. For each pair of participants, we compute the Jaccard similarity between their sets of state–label pairs, capturing the fraction of overlapping annotations relative to the total annotations made. In the setting *without additional guidance*, the average pairwise *Jaccard similarity is 0.1028*, indicating very high variability in how non‑experts describe the same trajectories. When we provide *additional guidelines* to the evaluators, the average *Jaccard similarity* increases to *0.2241*, so agreement between evaluators roughly doubles, although substantial variability remains. This analysis supports the reviewer’s point that human feedback can be highly noisy and underscores that our method is explicitly evaluated under such challenging, non‑expert conditions.
> > >
> > >     *Figure 4* in the main paper reports the error of the different algorithms (including PbRL with human evaluators) as a function of the number of human interactions, directly capturing the **number of interactions versus performance trade-off** in this non‑expert regime, as requested by the reviewer. Moreover, we note that the *average number of words per interaction is 17.48 and 17.30 words* in the settings without and with additional guidelines, respectively, providing a more granular view of the feedback cost in terms of annotation effort. We have added this information and the variability analysis to Appendix B.1.
> > >
> > > 5. We appreciate the reviewer's feedback about our claims. While our **claims** were intended to describe the specific experimental results presented in each section, we recognize they could be misinterpreted as broader generalizations. We now explicitly emphasize that all empirical claims are restricted to the Gridworld,  Reacher, and Rubik tasks. To ensure clarity and precision, we have revised the following statements (word changes and additions are in italics):
> > >     - RLHTF can rival access to privileged ground-truth rewards *in our experimental setting.*
> > >     - Here we prove the method’s robustness in settings that are both continuous and structurally complex $\rightarrow$ Here we *analyze the method’s performance in two* settings that are continuous and structurally complex
> > >     - *Towards addressing certain* reward modeling challenges
> > >     - As LLMs become more powerful, their ability to process human feedback improves, further boosting the performance of         RLHTF *in the Gridworld experiments.*
> > >     - RLHTF *in settings similar to those tested here may* continue to benefit from future advances in LLM capabilities.
> > >
> > >     These revisions maintain the integrity of our empirical findings while more precisely characterizing the scope of our contribution.

---

> > > > ### Comment · Reviewer_bg8B · 2025-12-09
> > > >
> > > > I thank the authors for their answers. Overall, I have not changed my opinion about the paper. In the rebuttal, the authors claim they prioritize interpretability over scalability, but a fair reading is that the simplicity and interpretability of the environments are necessary for the method to function at all. This is a much weaker claim than what is suggested in the narrative. As a result, in its current state the impact of the paper appears limited to such heavily structured, highly interpretable environments.
> > > >
> > > > I also note that the updated version of the manuscript contains a few typos:
> > > > 1. Section 5: “Goal Condition EL” should be “Goal-conditioned RL”.
> > > > 2. Section 6.2: “Towards Adressomg Certain” should be corrected (e.g., to “Towards Addressing Certain …”).

---

### Comment · Action_Editor_8tPW · 2025-11-26
**Authors-Reviewers Discussion Phase**

Dear Authors of Paper 6283,

As you may have seen in an email, we have now moved into the authors-reviewers discussion phase. Please take some time to process the reviewers comments and provide your response during the Authors-Reviewers discussion phase.

Dear Reviewers of Paper 6283,

Thank you for the time you put into reviewing the paper. Please read the other reviews to ensure you have all the information that is available prior to moving to the recommendation phase.

Best,

AE

---

### Decision · Action_Editor_8tPW · 2025-12-20

**Recommendation:** Accept with minor revision

**Additional Comments:**

The authors should ensure that their camera ready paper includes:
- all points raised by the reviewers that were addressed in the rebuttal. Specifically, please ensure that the claims made in the paper are limited to the environment studied in the paper
- Related to above, the AC observed that contribution 4 may be limited to only GridWorld environment. If true, please update the claim in the paper.

**Audience:**

Yes

**Audience Explanation:**

The paper provides a new approach to use prompt engineering for training reward models. This work should be of interest to members in the TMLR community that work on RLHF. This observation is supported by all three reviewers.

**Claims And Evidence:**

Yes

**Claims Explanation:**

The paper proposes a method named Reinforcement Learning from Human Text Feedback (RLHTF) that uses prompted language models to translate human-provided text to state-level rewards to train a reward model (RM). The reward model is then used to learn a policy in simple but interpretable environments (Gridworld, Reacher and Rubik's cube face). The key finding is that using guided free-form text allows for richer reward signal compared to preference information. A noteworthy contribution of this paper is that it provides a user study with 26 participants to validate findings with Gridworld environment.

All reviewers found the paper to be well written and exploring an interesting  direction. All reviewers agree that the paper provides sufficient evidence to support the claims made in the paper with the caveat that the environments used in the paper are somewhat simple. The authors explained their position in their rebuttal that the paper prioritizes small yet interpretable environments to allow for a careful study of the strengths and trade-offs with the proposed method.

Two of the three reviewers indicated that the claims made in the paper are supported by accurate, clear and convincing evidence while the remaining reviewer has concerns about the use of simple environments to support the claims. In the following, we discuss the important concerns raised during review, specifically the choice of environment and the challenges with prompt engineering:

## Use of simple environments
The use of Gridworld, a simple environment, allows for a meaningful study with 26 human participants as evaluators. This study allows the authors to study the core method in a real (or realistic) setting. An interesting finding from this study is that providing instructions to the evaluators allows them to effectively interact with the LLM in use to improve feedback. While we agree with the concern that the problem should be studied in a high dimensional setting, this should be noted as a limitation in the paper and left as topic for future work.

## Generalizing prompts to new environments
The paper's core approach with using prompt engineering to extract labels to train a reward model is direction that will be of interest to researchers in the RLHF community. The paper provides complete detail of the prompts used in their experiments in Appendix D. A concern here is that prompt engineering to new domains is a burden for any user that wants to extend this work to new environments. While designing prompts for new environments is indeed non-trivial, this concern does not invalidate the evidence provided in the paper as being insufficient.

## Outperforming baselines and ground truth rewards in certain cases
The paper lists the above as one of the contributions. While this claim is true for Gridworld (Figure 4), it is not clear whether the claim is true for other environments. The authors should clarify or adjust this claim in the paper.

In conclusion, the paper provides accurate, clear and convincing evidence to support the claims made in the paper and clarified during the Authors-Reviewers discussion.